

# Agricultural intensification vs climate change: What drives long-term changes of sediment load?

**Shengping Wang[1,2,3]\*, Peter Strauss[2], Carmen Krammer[2], Elmar Schmaltz[2], Borbala Szeles[3,4], Günter Blöschl[3,4]**

1.  *College of Hydraulic and Hydro-Power Engineering, North China Electric Power University, Beijing, 102206, P.R.China*

2.  *Institute for Land and Water Management Research, Federal Agency for Water Management, A-3252 Petzenkirchen, Austria*

3.  *Institute of Hydraulic Engineering and Water Resources Management, Vienna University of Technology, Vienna, Austria*

4.  *Vienna Doctoral Programme on Water Resource Systems*

**\*Corresponding author**: Shengping Wang

Email: wangshp418@126.com; Shengping_Wang@ncepu.edu.cn

**Abstract:** Climate change and agricultural intensification are expected to increase soil erosion and sediment production from arable land in many regions. However, so far, most studies have been based on short-term monitoring and/or modeling, making

it difficult to assess their reliability in terms of long-term changes. We present the results from a unique data set consisting of measurements of sediment loads from a 60ha catchment (the HOAL Petzenkirchen in Austria) over a time window spanning 72 years. Specifically, we compare Period I (1946-1954) and Period II (2002-2017) by fitting sediment rating curves for the growth and dormant seasons for each of the

periods. The results suggest a significant increase in sediment yield from Period I to Period II with an average of $11.6 \pm 10.8$ ton·yr$^{-1}$ to $63.6 \pm 84.0$ ton·yr$^{-1}$. The sediment

flux changed mainly due to a shift of the sediment rating curves (SRC), given that the

annual streamflow varied little between the periods (5.6 $l\,s^{-1}$ and 7.6 $l\,s^{-1}$, respectively,

on average). The slopes of the log regression lines of the SRC for the growing season

and the dormant season of Period I were 16.72 and 4.9, respectively, whilst they were

5.38 and 1.17 for Period II, respectively. Climate change, considered in terms of

rainfall erosivity, was not responsible for this shift, given that erosivity decreased by

30.4% from the dormant season of Period I to that of Period II, and no significant

difference was found between the growing seasons of Periods I and II. However, the

sediment flux changes can be explained by changes in crop type and parcel structure.

During low and median streamflow conditions (*i.e.* $Q < Q_{20\%}$), land consolidation in

Period II (*i.e.* theparcel effect) did not exert an apparent influence on sediment

production. Whilst with increasing stream flow ($Q > Q_{20\%}$), parcel structure played an

increasingly role in sediment yield contribution, and leading to a dominant role due to

enhanced sediment connectivity in the landscape at extremely high flow conditions

(*i.e.* $Q > Q_{2\%}$). The increase in cropland in Period II at the expense of grassland had an

unfavourable effect on sediment flux, independent of streamflow, with declining

relevance as flow increased. We conclude that both land cover change and land

consolidation should be accounted for simultaneously when assessing sediment flux

changes. Especially during extremely high flow conditions, land consolidation

substantially alters sediment fluxes, which is most relevant for long-term sediment

loads and land degradation. Increased attention to improving parcel structure is

therefore needed in climate adaptation and agricultural catchment management.



**Keywords: Sediment regime; Land use/cover change; Parcel structure;**

**Climate change; Agricultural catchment**

## Introduction

Soil erosion is a risk of worldwide importance because of its environmental and

economic consequences (García-Ruiz, 2010; Prosdocimi et al., 2016). Climate change,

land use/cover changes and other anthropogenic activities are commonly considered

potential agents driving variation of soil erosion rates (Nearing et al., 2004;

Jean-Baptiste et al., 2015; Zhang et al., 2021). The impacts of future climate

projections or recently observed climate change on soil erosion have been explored in

a number of studies (Nearing et al., 2004; Zhang and Nearing., 2005; Mullan, 2013;

Jean-Baptiste et al., 2015; Palazon and Navas, 2016; Mullan et al., 2019). Changes in

land use or land cover have also been widely investigated, using either experimental

observations or modelling approaches at various scales (e.g. Korkanç et al., 2018;

Silva et al., 2017; Bochet et al., 2006; Karvonen et al., 1999; Ozsahin et al., 2018;

Nampak et al., 2018; Li et al., 2019; Perović et al., 2018). Additionally, the relative

contributions of climate change and land use/cover change have been increasingly

investigated in recent years as both agents usually exert their influence on soil erosion

simultaneously. In a review on anthropogenic and climatic impacts on Holocene

sediment dynamics of the Western Mediterranean basin (Bellin et al., 2013), the

impacts of climatic and anthropogenic activities on sediment dynamic were found to





interact with each other during moist climatic periods, whilst the enhanced sediment

load was found only closely associated with enhanced aridity during dry climatic

periods, independent of the intensity and type of human activities. By using field

investigation combined with modeling, Zhang et al. (2021) quantitatively evaluated

the contribution of climate change, land use, and silt trap dams to sediment reduction

of a typical Loess watershed over 1987-2016, with contribution values being 29%,

40%, and 31%, respectively. Sun et al. (2020) quantified the contribution of climate

change and land-use change to sediment reduction of a Loess watershed over 1997 to

2016, in which both engineering measures and land-use change accounted for 62%

and 23 - 42% of sediment reduction, respectively, the rest being explained by climate

change. Management practices generated a higher impact on soil erosion at the plot

scale than climate change (Scholz et al., 2008). Also, livestock grazing accelerated

soil erosion more than climate change in Qinghai-Tibet Plateau (Li et al., 2019).

The previous findings provide valuable information for implementing water and soil

conservation measures and improving agricultural productivities (Nampak et al., 2018;

Li et al., 2019). However, the role of landscape structure changes has so far not

received much attention, even though land-use policies such as land consolidation

have been changing agricultural practices to a large extent since the beginning of

agricultural industrialization after 1945 (e.g. Moravcová et al., 2017; Devaty et al.,

2019) and in particular in countries where the industrialization of agriculture is

relatively recent (Bouma et al., 1998; Moravcova et al., 2017; Zhang et al., 2021).

Landscape structures usually influence erosion due to the boundary effects between



land uses and land units (parcels) that differ in water and sediment trapping capacity (Baudry and Merriam, 1988; Merriam, 1990; Takken et al., 1999; Phillips et al., 2011).

Van Oost et al. (2000) and Devaty et al. (2019) evaluated the role of landscape structure by accounting for its spatial connectivity using modelling approaches and found that landscape structure is an essential factor when assessing the risk of soil erosion affected by land-use changes. Both studies emphasized the potential impacts of parcel structure changes on sediment production through altering hydrological and

sediment connectivity. However, both authors relied on models, making connectivity assumptions in their studies. Instead of focusing on the spatial connectivity, others (e.g. Bakker et al., 2008; Sharma et al., 2011; Chevigny et al., 2014;Wang et al., 2021; Tang et al., 2021; Madarász et al., 2021) evaluated the effect of terrain, soil properties, lithology, management practices and other processes associated with landscape and/or

land structure changes and highlighted their impact on sediment production. It has also been shown that the impact of landscape structure on erosion is more heterogeneous when different crops are grown, and the underlying lithology, soil properties and topography show substantial spatial variability across the catchment (David et al.,2014; Cantreul et al., 2020).

Even though numerous studies have addressed the effect of climate change and land management on soil erosion and sediment production, most studies have been based on short-term monitoring and/or modeling, which makes it difficult to assess their reliability in terms of long-term changes that are the most relevant from a practical perspective. This paper aims at evaluating the relative roles of climate change, land



use and land cover changes (LUCC) and change of land structure on sediment

production. We present the results from a unique data set consisting of measurements

of sediment loads from a small agricultural catchment over a time window spanning

72 years. The catchment is the 66 ha Hydrological Open Air Laboratory (HOAL)

Petzenkirchen (Blöschl et al., 2016), which, in addition to being exposed to climate

change, has experienced a significant change in land use and land cover as well as

parcel structure for erosion control during the past decades. Both discharge and

sediment yield have been monitored in the HOAL catchment since the 1940s. This

provides an opportunity for disentangling the impact of parcel structure and land

use/land cover change, and the impact of climate change based on long-term

measurements. Specifically, we aim at i) exploring how the sediment regime has

changed between the periods of 1946-1954 and 2002-2017; ii) analyzing whether

climate change or land-use changes (or both) were responsible for any change in

sediment regime; and iii) identifying the relevance of land structure change (i.e. land

consolidation) on erosion control compared to that of a change in land use or cover.

## 2. Methods

### 2.1 Catchment characteristics

The HOAL catchment is situated in Lower Austria's alpine forelands (48°9' N, 15°9' E)

with elevations ranging from 268 m to 323 m a.s.l. and a size of 66 ha. The climate of

the catchment belongs to the temperate, continental climate zone (Dfb) according to

Köppen-Geiger (Kottek et al., 2006), with a mean annual precipitation of 746 mm

(1946 - 2006), 62% of the rain falling between May and October. The mean daily air





temperature is 8.8°C (1946-2006), and the dominant land use is arable land, accounting for, on average, 82% of the catchment over the past few years. Typical crops include winter wheat, corn and barley. Deciduous trees grow along the creek

(6%), 10% of the area is grassland, and 2% is paved. The subsurface of the catchment consists of tertiary marine sediments. Soils are classified into five types: calcic cambisols, vertic cambisols, gleyic cambisols, planosols and gleysols (IUSS Working Group WRB, 2015).

## 2.2 Data availability


Both streamflow discharge ($Q$, l/s) and sediment concentration ($C$, mg/l) have been measured at the catchment outlet since the 1940s. Data records from 1946-1954 (Period I) and 2002 to 2017 (Period II) were available for this analysis. Due to technological advances, the data was measured with different methods. In Period I,

discharge was registered at 10 min resolution by a Thompson weir and a paper chart recorder, while in Period II, it was registered at 5 min resolution by an H-Flume and a pressure transducer. Sediment concentrations were measured manually every 3-4 days in Period I, whilst an automatic method plus additional manual sampling was applied in Period II. Daily precipitation and 5-min rainfall intensities were available for both

periods, but for Period I, 5-min rainfall intensities were only available during the vegetation period.

We used parcel-based land use data from 1946 to 1949 and 2007 to 2012, representing Period I and Period II land use, respectively. Land use categories were agricultural land, including crop type, grassland, forest, roads and settlements. We defined a parcel

as a continuous area of land with a single crop type. Parcel boundaries were specified

according to the cadastral map and aerial photographs. In Period II, these boundaries

were also visually inspected. Figure 1 depicts the geographic catchment location, and

parcel structure and land use for a specific year in each period.

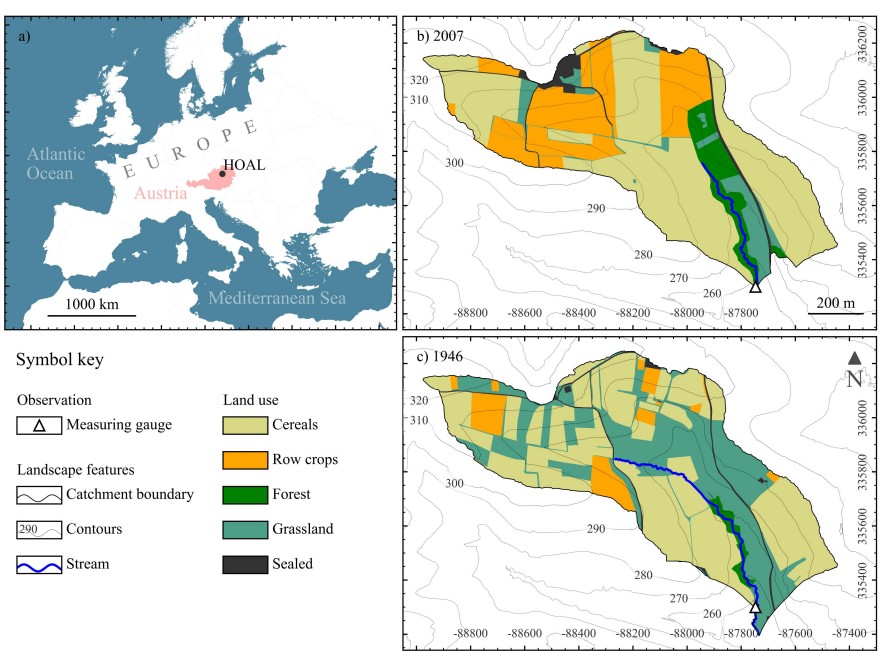


Figure 1 Geographical location of the HOAL catchment in Petzenkirchen in Austria
and Europe and Parcel structure and land use in the HOAL catchment for 1947 (a) and
2007 (b). The black hatched area in b) represents a difference in catchment size due to
the relocation of the stream gauge in Period II. Coordinates as EPSG: 31256 – MGI /
Austria GK East (meters).

## 2.3 Data analysis

### 2.3.1 Changes in rainfall erosivity and flow regime

The effect of rainfall on erosion was quantified by the R-factor of the Revised

Universal Soil Loss Equation (RUSLE), which is defined as the product of kinetic



energy of a rainfall event and its maximum 30-min intensity, using the rainfall

erosivity tool RIST (USDA-Staff, 2019) according to

$$EI_{30} = \sum_{i=1}^{m} E_i \cdot I_{30,i} \qquad (1)$$

where $EI_{30}$ is the Annual R-factor (N·h$^{-1}$·yr$^{-1}$) calculated as the sum of single event

R-factors, $E_i$ is the total kinetic energy for a single event (kJ·m$^{-2}$), $I_{30}$ is the maximum

rainfall intensity in 30 minutes within a single event $i$ (mm·h$^{-1}$), and $m$ is the number

of events per year.

We assumed erosivity density $ED$ (*i.e.* $EI_{30}$ divided by event precipitation) to be a

particularly relevant climatic indicator of the soil erosion process and catchment

sediment yield. We, therefore, tested whether the means of the monthly erosivity

density ($ED$) are significantly different between Period I and Period II by using a

t-test. Due to the absence of rainfall intensity measurements, we could not calculate

$ED$ for the months of the dormant season (November to March) of Period I. For this

case, we calculated $ED$ from a relationship between $EI_{30}$ and monthly rainfall of

Period II, assuming that the relationship was sufficiently temporally invariant over the

investigated periods. Erosivity is very low during the dormant season (Figure 3a); thus,

the error arising from the use of this relationship is expected to be small.

We also compared daily flow duration curves to understand whether hydrological

regime change has influenced flow transporting capacity and sediment regime

variation. Following the definitions of Smakhtin (2001), we compared low flow

($Q_{70\%}$), high flow ($Q_{10\%}$) and median flow rate ($Q_{50\%}$) quantiles for the two periods.

**2.3.2 Sediment regime analysis**

Sediment regimes were mainly analyzed using sediment rating curves (SRC).

Following a common approach (Asselman, 2000; Warrick and Rubin, 2007; Sheridan

et al., 2011; Vaughan et al., 2017; Khaledian et al., 2017), the SRCs were assumed to

follow a power-law function, which was fitted by least squares regression:

$$C = a \cdot Q^b \qquad\qquad (2)$$

where $C$ is sediment concentration ($mg\,l^{-1}$), $Q$ is discharge ($l\,s^{-1}$), and $a$ and $b$ are

dimensionless regression coefficients. The coefficient $a$ is usually associated with

easily transported intensively weathered materials and may vary over seven orders of

magnitude (Syvitski et al., 2000). The parameter $b$ represents the capacity of the

stream to erode and transport sediment, reflecting how sediment concentration is

non-linearly related to streamflow (Sheridan et al., 2011; Fan et al., 2012). It typically

varies from 0.5 to 1.5 and rarely exceeds 2. Sometimes $b$ is also regarded as a measure

of the quantity of new sediment sources available (Vanmaercke et al., 2010; Guzman

et al., 2013).

Considering that data records were registered with different resolutions for Periods I

and II (See section 2.2), for the sake of consistency, we used monthly averages, as

conducted in the other studies (Syvitski and Alcott,1995; Sheridan et al., 2011; Hu et

al., 2011), to construct SRC. We assumed that monthly averages could reflect a varied

hydrological and/or sediment response to seasonally prevailing weather characteristics

such as dry periods or convective storms (Sheridan et al., 2011).

We chose arithmetic means of the observations to represent the monthly $Q$ and $C$

values. These monthly averages were pooled together and then grouped into growing





season of Period I (Period I_G), dormant season of Period I (Period I_D), growing

season of Period II (Period II_G), and dormant season of Period II (Period II_D),

respectively. For each of these four periods, we fitted SRC.

We analyzed the fitted SRC by two strategies to evaluate whether and how the

sediment regime changed between these periods. Besides directly comparing the slope

of the four seasonal SRC, we also fitted SRC by season and year and plotted the

regression coefficients *a* against their corresponding *b* to evaluate a possible sediment

regime shift during Periods I and II. Thomas (1988) suggested that this technique

could exclude the interference of different sampling practices with the estimated

sediment regime.

It is suggested that for the years (or catchments) with similar means of log-Q and

log-C, SRC would usually pass through one common point O (Thomas, 1988;

Syvitski et al.,2000; Desilets et al., 2007; Sheridan et al., 2011). This common point O

(Figure 2a) is usually interpreted to reflect time invariant catchment characteristics,

such as relief, drainage area, and drainage density, while the variation of the slope of

SRCs (Figure 2a) is interpreted to reflect temporally dynamic characteristics, such as

average or maximum discharge and sediment availability (Asselman, 2000). The

coefficients *a* of the SRCs having a common point are usually inversely linearly

related with *b* as well (Thomas, 1988, Syvitski et al., 2000 and Desilets et al., 2007).

This provides a means for testing whether periods (or catchments) have similar

sediment transporting regimes by plotting coefficient *a* against *b* (Figure 2b). That is

to say, thatthe points plotted on the same line (A, B, C) in Figure 2b are representative



of periods (or catchments) having similar sediment transport regimes. Points A of

Figure 2b (upper-left-side) usually exhibit steeper rating curves than points C

(lower-right side). For different lines in Figure 2b, the lower ones represent situations

with most of the annual sediment load transported at relatively low flow discharge,

and the higher ones represent situations with suspended sediment mainly transported

at high streamflow. Thus, it is possible to reveal changes in sediment transport

regimes. Compared to a direct evaluation of rating curves, relating coefficient $a$ to

exponent $b$ is more conductive to revealing the temporal evolution of the sediment

regime (Syvitski et al., 2000; Desilets et al., 2007).

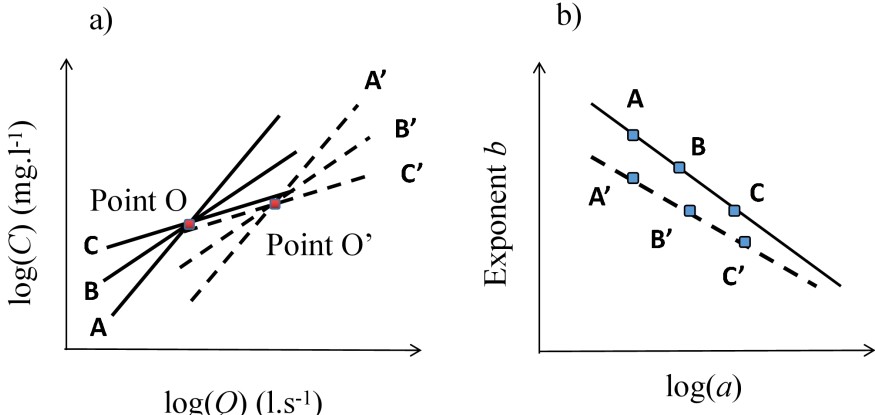


Figure 2 Schematic showing a) how sediment rating curves (SRC) may rotate around
a common point and b) how exponents $b$ of the SRC relate to coefficients $a$. Lines A,
B and C on the left are SRC of different periods (e.g. years) sharing a similar common
point O. Once sediment regimes shift due to the changes in catchment characteristics
(change in drainage density, drainage area, and topography…..) the common point O
would change to point O', and the linear relationship between $a$ and $b$ of the SRC
would exhibit a shift as well. The schematic is based on $\log C = \log a + b \cdot \log Q$
(Equation 2).





To account for uncertainties of the fitted SRC during each period, we additionally

established theoretical sediment rating curves (tSRC); i) for each period (*i.e.* Period

I_G, Period I_D, Period II_G, and Period II_D), we carried out random sampling of

*log a* (n=500, package "sample" in RStudio), assuming that the samples of the

coefficient of log *a* follow normal distributions (Figure 4), which was proved with a

Kolmogorov-Smirnov test of normality (mean = 1.02, SD = 2.01, n=44); ii)

given the set of the sampled 500 values of log *a*, we generated a set of

values*b*according to the previously established linear relationship between log *a* and *b*;

iii)      given a set of specified Q values, we derived 500 tSRC for each period,

corresponding to the paired log *a* and *b* samples; iv) using these tSRC we calculated

the 50 percentile, 5 percentile, and 95 percentile for each period to reveal uncertainties

of the sediment rating curves.

     The tSRC of the periods were also used to quantify the effect of land consolidation, *i.e.*

change of parcels structure and sizes (Parcel_effect) versus the effect of land use and

land cover changes (LUCC_effect).Since vegetation usually plays a minor role in the

dormant season due to the absence of a dense vegetation cover on arable land and

little erosive rainfall (Madsen et al., 2014; Kundzewicz, 2012; Salesa and Cerda, 2020;

Hou et al., 2020), landscape structure in the dormant season was considered a

critically important factor affecting runoff production and sediment production

(Sharma et al.,2011; Devátý et al., 2019). Therefore, we hypothesized that the total

change in sediment yield (Total effect) resulted from land cover change





(LUCC_effect), land structure change (Parcel_effect) and climate change

(Climate_effect). The effects of land cover and land structure change could be

quantitatively separated according to the seasonal differences in tSRC after

determining the climate change effect. Specifically, we assume that the shift of

sediment regime from Period I_D to Period II_G is representative of the Total_effect

(Equation 3), and the shift in sediment regime between Period I_D and Period II_D is

mainly due to land consolidation (Parcel_effect) (Equation 4). Thus, the LUCC effect

could be estimated according to Equation (5) if the Climate_effect was insignificant

(section 3.1). The contributions of Parcel_effect and LUCC_effect to the Total_effect

were estimated according to Equations (6) and (7), respectively.

$$\text{Total\_effect} = tSRC_{50\%}(\text{Period II\_G}) - tSRC_{50\%}(\text{Period I\_D}) \qquad (3)$$

$$\text{Parcel\_effect} = tSRC_{50\%}(\text{Period II\_D}) - tSRC_{50\%}(\text{Period I\_D}) \qquad (4)$$

$$\text{LUCC\_effect} = \text{Total\_effect} - \text{Parcel\_effect} - \text{Climate\_effect} \qquad (5)$$

$$\text{Parcel\_effect (\%)} = \frac{\text{Parcel\_effect}}{\text{Total\_effect}} \times 100 \qquad (6)$$

$$\text{LUCC\_effect (\%)} = \frac{\text{LUCC\_effect}}{\text{Total\_effect}} \times 100 \qquad (7)$$

## 3. Results

### 3.1 Changes in climate and flow regime

Because climate is commonly found responsible for hydrological change (e.g., Kelly

et al., 2016; Wang et al., 2020), we compared erosivity density (*ED*) and monthly



precipitation (P) of the two periods to examine whether climate affected the variation of sediment regime in the catchment. The mean monthly *EDs* in the growing season were $2.37 \pm 1.38$ and $1.84 \pm 0.86$ (N·h⁻¹·yr⁻¹·mm⁻¹) (standard deviation between years),

respectively, which was not significantly different ($p>0.05$) between the first and second period (Figure 3a). In contrast, mean monthly *ED* in the dormant seasons showed a significant ($p<0.05$) decrease from the first to the second period ($0.66 \pm 0.21$ and $0.42 \pm 0.11$ (N·h⁻¹·yr⁻¹·mm⁻¹), respectively. No significant difference was found between the first and second periods for the mean monthly P in dormant or growing

season (Figure 3b). The mean monthly *P* in Period I was $50 \pm 33$ and $76 \pm 54$ mm for the dormant and growing season, and $53 \pm 29$ mm and $79 \pm 47$ mm in Period II. The decrease in *ED* during the dormant season of Period II and the insignificant change in monthly P suggest that climate change between Period I and II was not responsible for an increased sediment availability (see section 3.3).

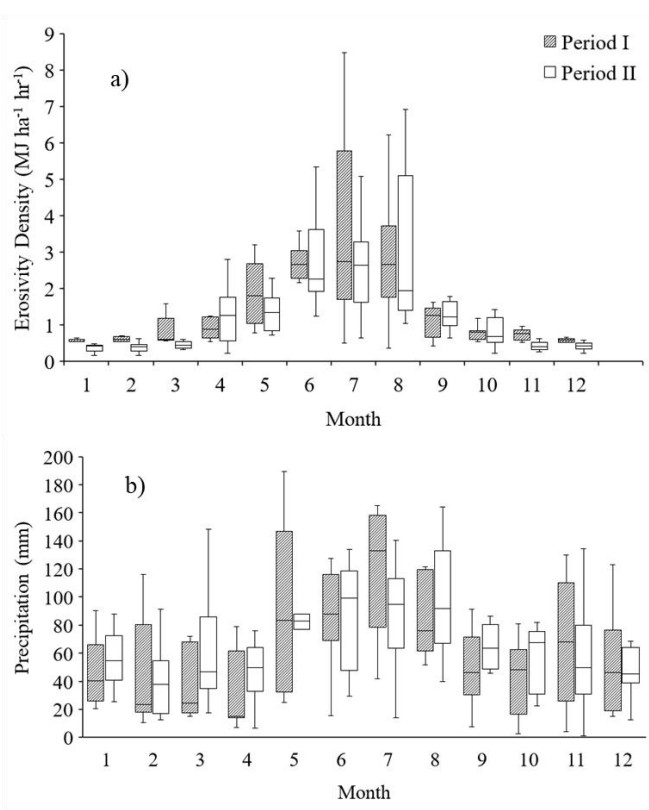


Figure 3 Distribution of a) monthly mean erosivity density and b) monthly precipitation for Periods I and II.

Daily flow duration curves for both periods are displayed in Figure 4. Generally, daily

streamflow in Period I was higher than that of Period II. The $Q_{70\%}$ low flow of the two

periods was 2.3 and 1.9 $l\,s^{-1}$, the $Q_{50\%}$ median flow was 3.1 and 2.7 $l\,s^{-1}$, and the $Q_{10\%}$

high flow was 7.3 and 6.5 $l.s^{-1}$, respectively. The decreased flow regime of Period II,

probably in part due to increased evapotranspiration over the past decades

(Duethmann and Blöschl, 2018), suggests a smaller streamflow transport capacity and

indicates that it was not responsible for the increased sediment transport in Period II

(see section 3.3).

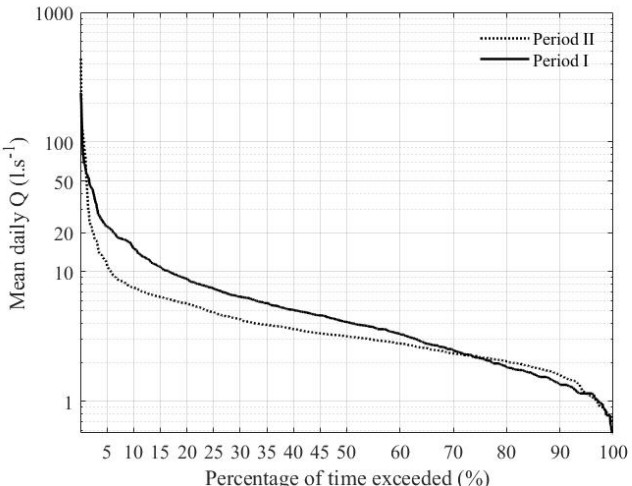

Figure 4 Mean daily flow duration curve for Periods I and II.


## 3.2 Change in land use and land organization

Table 1 shows how land use changed between the two periods. During Period I,

cropland and grassland accounted for 73% to 82% and 14% to 22% of the area.

However, due to agricultural intensification, cropland increased to around 82% in

Period II, at the expense of a decreasing share of grassland. Forest, including sparse

forest, accounted for 1.8% area during Period I but increased considerably until

Period II to around 11%. Within the land use class of arable land, a substantial change

from crops with low risk to cause soil erosion to crops with a high soil loss potential

appeared. This was particularly true for maize. In addition, the diversity of crops

decreased considerably (Table 2). This shift to agricultural uniformity is likely to

affect also land structure effects.



Besides the change in land use, the parcel structure of the catchment also changed

(Table 1). This change was related to a land consolidation plan issued in 1955 in

Austria (Deváty et al., 2019) and a massive trend to agricultural industrialization that

evolved after 1945. During Period I, arable land was fragmented across many small

parcels, with a mean parcel size between 0.5 - 0.6 ha and a parcel density (number of

parcels per ha area) between 1.7 - 2.0 ha$^{-1}$ in different years. In Period II, these values

increased considerably to mean parcel sizes between 1.7 - 2.7 ha and parcel densities

between 0.3 - 0.6 ha$^{-1}$. Similarly, the mean parcel size and parcel density of grassland

during Period I were 0.13 - 0.17 ha and 5.2 - 7.2 ha$^{-1}$. It had changed to 1.06 ha and

0.9 ha$^{-1}$ in Period II.

**Table 1 Parcel and land use statistics for Periods I and II. Land use for the years 1946 to**
**1949 represents Period I, land use for the years 2007 to 2012 represents Period II; N is the**
**number of parcels for a given land use, density is the number of parcels per ha, mean size**
**represents the mean area of parcels with a particular land use.**

| | Parcel Structure | | | | | | | |
|---|---|---|---|---|---|---|---|---|
| Land use | Period I | | | | Period II | | | |
| | N | Density (1·ha$^{-1}$) | Mean size (ha) | Area (%) | N | Density (1·ha$^{-1}$) | Mean size (ha) | Area (%) |
| Arable land | 70-111* | 1.7-2.0 | 0.5-0.6 | 73-82 | 21-33 | 0.3-0.6 | 1.7-2.7 | 81-82 |
| Grassland | 70-81 | 5.2-7.2 | 0.1-0.2 | 14-22 | 6 | 0.9 | 1.1 | 3-4 |
| Forest | 1 | - | 1.2 | 1.8 | 7 | 1 | 1.0 | 10.5-11 |
| Paved area | 17 | 12.9 | 0.1 | 2 | 17 | 7.3 | 0.1 | 2.4 |

* The number of parcels varied with the specific year of a period


**Table 2 Crop statistics of arable land for Periods I and II; Crop statistics for the years 1946**
**to 1949 represent Period I, crop statistics for the years 2007 to 2012 represent Period II;**
**Erosion risk for a particular crop is classified as high or low according to the classification of**
**management in the RUSLE.**


| | Period I | Period II |
|---|---|---|





| | Area (ha) | Area (%) | Area (ha) | Area (%) | Erosion risk |
|---|---|---|---|---|---|
| Meadow | 9-15 | 18-30 | 0.8 | 2 | low |
| Alfalfa | 11-18 | 22-33 | - | - | low |
| Wheat | 5-14 | 9-26 | 3-35 | 5-66 | low |
| Rye | 3-13 | 5-24 | | | low |
| Beets | 2-12 | 3-22 | - | - | high |
| Oats | 2-10 | 4-18 | 2 | 4 | low |
| Barley | 0.3-8 | 5-15 | 2-29 | 5-55 | low |
| Potatoes | 3-7 | 6-14 | - | - | high |
| Maize | 0.3-0.8 | 0.6-1.1 | 6.3-34 | 12-63 | high |
| Rape | - | - | 0.7-23 | 1-43 | low |
| Sunflower | - | - | 0.2-2 | 0.3-4 | high |

## 3.3 Change in sediment transport regime

### 3.3.1 Direct comparison of the fitted SRCs

Figure 5 shows the fitted sediment rating curves ($p<0.05$) for both periods. Although

rainfall erosivity of Period II_G was similar to that of Period I_G (Figure 3a) and

streamflow of Period II was generally lower than that of Period I (Figure 4), the fitted

SRC of Period II_G was steeper than that of Period I_G (Figure 5a), with the

coefficients $b$ being 0.32 and 1.65for Period I_G and Period II_G, respectively (Table

3). The rating curves of the dormant seasons demonstrated a faster response of

sediment concentration to increasing flow in Period II_D (Figure 5b), the coefficients

$b$ being 0.75 and 1.69 for Period I_D and Period II_D, respectively. However, the

rainfall $ED$ in Period II_D was generally lower than that of Period I_D (Figure 3a),

suggesting a lower probability of a substantial increase in sediment availability. These

results indicate that neither changes in rainfall erosivity nor the hydrological regime

could explain the increase in sediment dynamics.



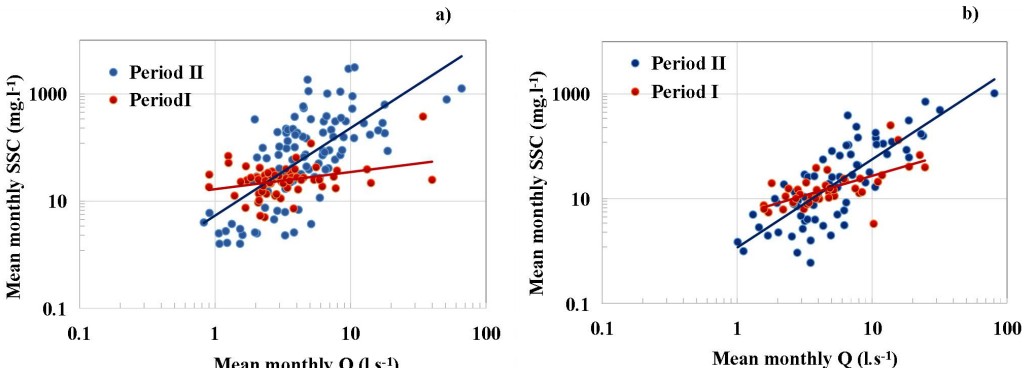

Figure 5 Sediment rating curves for a) the growing seasons and b) the dormant seasons in the two periods. Each point represents one mean monthly observation.


**Table 3 Parameter values for the coefficients of the SRC for different periods and seasons according to Equation (2).**

| Period | Coefficient | | $r^2$ |
| --- | --- | --- | --- |
| | *a* | *b* | |
| Period I_G | 16.72 | 0.32 | 0.11 |
| Period I_D | 4.90 | 0.75 | 0.42 |
| Period II_G | 5.38 | 1.63 | 0.45 |
| Period II_D | 1.17 | 1.69 | 0.64 |

### 3.3.2 Relationship between coefficient *a* and *b*

The changing steepness of a fitted SRC does not necessarily imply a change in

sediment regime as slopes of fitted SRC sometimes are affected by catchment size or

the distribution of samples (Asselman, 2000). To minimize possible interference of

other factors in identifying variation or shift of SRC, we investigated the relationship

between coefficients *a* and *b* of the SRC. This technique was used by Asselman et al.

(2000) and Fan et al. (2012) to examine the differences in sediment regimes between

spatially different sites. Additionally, Sheridan et al. (2011) used the relationship to

reveal post-fire temporal shifts of a sediment regime.

Figure 6 displays the coefficients log *(a)* plotted against *b* for the four investigated

time frames. Even though monthly averages were used for the sake of consistency, the

different sampling strategies of Periods I and II caused a biased distribution of log *(a)*

and *b*. Data of Period I is concentrated in the right-lower area (blue points). In contrast,

data of Period II is more concentrated in the left-upper area, which is especially true

during the dormant season.

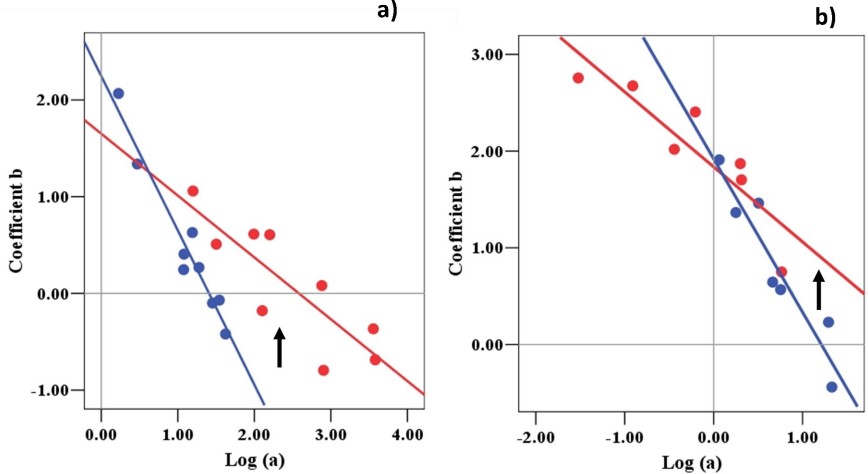


Figure 6 Relationship between coefficients *a* and *b* for a) growing season and b)
dormant season of Period I (blue) and Period II (red), respectively. All regressions are
significant at p<0.05.

Nevertheless, it is evident that the regression line exhibits a shift between the periods,

the slopes of the regression line changing from -1.60 to -0.94 in the growing season

and from -1.58 to -0.80 in the dormant season (Figure 6). The shift of the line to the


upper direction at log (a) larger than around 0.6 suggests that in Period II, most of the

sediment was transported at relatively high flow rates. Since climate change was not

responsible for the increased hydrological regime (see section 3.1), we mainly

attribute this shift to the increase in hydrological connectivity, such as flow path

density and flow length, and a change in land use and land cover statistics.

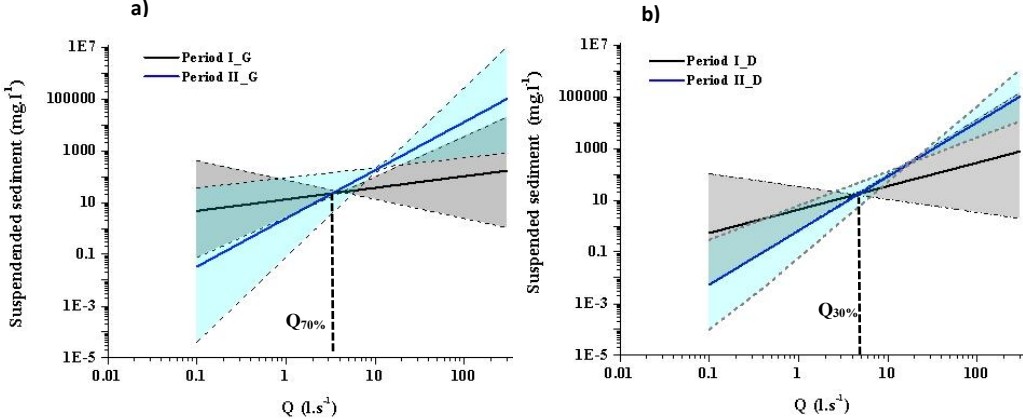

Figure 7 tSRC for the growing seasons (a) and the dormant seasons (b) of Period I and
II. Solid lines denote the 50 percentile of the tSRC for each period. The grey area
denotes the range of the predicted tSRC composed of 5 and 95 percentiles. $Q_{30\%}$ and
$Q_{70\%}$ and represent the flow conditions of 3.9 $l \cdot s^{-1}$ and 2.0 $l \cdot s^{-1}$, respectively.

Figure 7 displays the tSRC with their uncertainties for the different periods and

seasons, to allow for directly discriminating the change in sediment regime with

uncertainty. During most of the time, *i.e.* at flow rates larger than approximately $Q_{70\%}$,

sediment concentrations for a given Q in Period II_G were considerably higher than in

Period I_G (Figure 7a), whilst at flow rates below this value, sediment concentrations

were not different. This finding is supported by the significant change in the sediment





load of $6.3 \pm 19.9$ ton per month during Period II_G compared to $0.8 \pm 3.3$ ton per

month during Period I_G. Sediment concentration in the dormant season of Period II

was considerably higher than that of Period I at flow rate exceeding $Q_{30\%}$ (Figure 7b).

Again, this confirms the shift of sediment transport regime in Period II_D and is in

line with the increase in sediment load in Period II_D, resulting in mean monthly

sediment loads of $5.4 \pm 18.3$ ton per month compared to $1.3 \pm 3.9$ ton for Period I_D.

## 3.4 Parcel_effect versus LUCC_effect

Figure 8 demonstrates the dynamic contributions of land structure and land cover

changes on sediment concentrations with increasing flow. Land consolidation and the

substantial increase in cropland at the expense of a decrease in grassland explained the

increase in sediment yield. However, the trends of their contributions to this increase

differed. Generally, with higher flow rates, the contribution of the LUCC_effect

gradually decreased, whilst the contribution of the Parcel_effect increased. The

Parcel_effect accounted for more than 50% of the Total_effect after the flow rate

exceeded 20 $l\,s^{-1}$ approximately (*i.e.* $Q_{2\%}$) (Figure 8), exhibiting a dominant role in

sediment production. The opposite trend of the contributions between LUCC_effect

and Parcel_effect suggests that, even though land consolidation and an increase in

cropland both have unbeneficial effects on erosion control, their hydrological

consequences may be different, with land structure change probably explaining much

of the variation of sediment load at high flow conditions.

Unlike the situation during high flow rates, the Total_effect showed an almost zero



value at flow rates less than approximately 2 $l\,s^{-1}$ (*i.e.* $Q_{70\%}$) (Figure 8), suggesting no

difference in sediment load between Periods I and II at low flow conditions. The

increase in sediment load, at flow rates of 2 up to around 20 $l.s^{-1}$, seemed mainly due

to the increase in the cropland of Period II, as the contribution from LUCC_effect was

consistently higher than that of the Parcel_effect.

One may note that forest cover increased considerably from Period I to Period II. It,

however, did not show an influential role in erosion control. We hypothesize that even

though a beneficial effect of forest increase (accounting for 11% around of the

catchment) may have appeared in Period II, it was offset by the negative effect of crop

land changes, particularly the increase in erosive row crops, which contribute

substantially to sediment yield compared with other land uses and other crop types

(Kijowska‑Strugała et al., 2018).

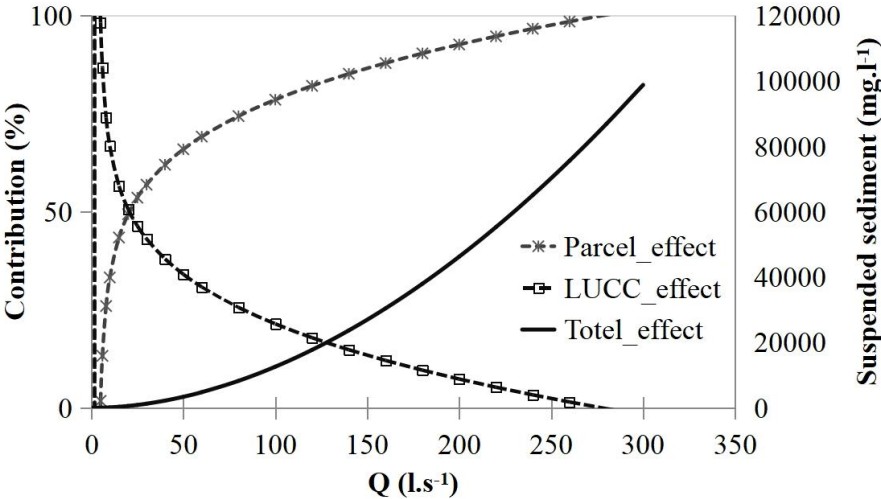


Figure 8 Contribution of Parcel_effect and LUCC_effect to the Total_effect across
various flow rates. Total_effect (Equation 3) is displayed in terms of suspended
sediment concentration. Parcel_effect and LUCC_effect was estimated by Equations



(4) and (5), respectively; their contribution to the Total_effect was estimated by
Equations (6) and (7), respectively.

## 4. Discussion

Industrial intensification of agriculture implemented in the last 70 years has raised

considerable concern regarding erosion and sediment loading of rivers (e.g. Bakker et

al., 2008; Chevigny et al., 2014). However, with global climate warming, the different

contributions to sediment load from land use and land cover change, land policy

adjustments such as land consolidation and climate change are not well understood.

This paper aims at evaluating the relative roles of climate change, land use and land

cover changes, and land consolidation in sediment production, particularly for varying

flow rates.

We found that sediment load increased substantially from Period II to Period I.

Climate change in terms of both monthly $ED$ and $P$ was not responsible for this effect,

instead it can be explained by land cover and land consolidation changes. Their

relative contributions varied with streamflow. For flow conditions below around 5

$l.s^{-1}$($i.e.$ $Q_{20}$%), land consolidation had no apparent adverse effect on erosion control,

but with increasing flow, the contribution to sediment load increased continuously,

leading to a dominant role at extremely high flow rates. This finding is partially in

line with David et al. (2014) and Cantreul et al. (2020). They reported that landscape

structure was less important for soil erosion during most normal flow conditions than

land use and land cover. However, they did not investigate whether the effect of

landscape structure showed a dynamic behavior with increasing flow. In contrast, the





LUCC_effect, i.e. the increase of crops with high erosion risk, continuously affected

sediment load with gradually decreasing importance for high and extreme flow

conditions. Similar results have been reported by Vaughan et al. (2017), who showed

that sediment concentration at low and median flow conditions was considerably

associated with a change in catchment characteristics, primarily land use and land

cover.

Although the effect of land use changes was dominating for flow conditions below

$Q_{20}$%, it's contribution to the total annual sediment load was small. More than 75% of

the total sediment load was transported during a small number of events (25 events in

Period I, 8 events in Period II) and all events had flow rates above 15 $l.s^{-1}$., which

underlines the importance of land structure for sediment loading.

This behavior is associated with the processes and mechanisms of vegetation

controlling overland flow as a transporting agent for sediment (e.g. Sun et al., 2013;

El Kateb et al., 2013; Nearing et al., 2017; Kijowska‑Strugała et al., 2018; Silasari et

al., 2017). A change in land use and land cover implies alterations of surface

characteristics, such as above ground structure morphology, litter cover, organic

matter components, root network (Gyssels et al., 2005; Wei et al., 2007; Moghadam et

al., 2015; Patin et al., 2018) and soil properties (Costa et al., 2003; Moghadam et al.,

2015). These properties influence the protective role of vegetation in soil detachment,

flow capacity to transport sediment particles, and runoff flow paths to river channels

(Van Rompaey et al., 2002; Lana-Renault et al., 2011; Sun et al., 2018). Nevertheless,





the protective effects do not linearly increase with increasing surface runoff.

Accelerated discharge and stronger scouring effects of upslope discharge might impair

the protective role of vegetation (*e.g.* Zhang et al., 2011; Santos et al.,2017; Yao et al.,

2018; Bagagiolo et al., 2018; Wang et al., 2019). Vegetation usually exhibits a smaller

interception capacity at high rainfall intensity, resulting in enhanced splash erosion

and availability of mobile soil particles (Cayuela et al., 2018; Magliano. et al., 2019;

Nytch et al., 2019). However, the decreasing contribution of the LUCC_effect does

not directly imply an absolute decrease of the magnitude of the LUCC_effect. The

absolute change in SSC resulting from LUCC reveals an increasing trend as flow rates

increases. Thus, the contribution of the LUCC_effect stands for the relevance of

LUCC in erosion control compared to the change due to land consolidation. The

magnitude of the LUCC_effect probably depends mainly on where within the

catchment the land cover was replaced and how the proportional area of various land

uses changed. We will address this issue in future analyses.

Unlike land cover and land use change, landscape structure is usually combined with

other catchment properties, such as slope characteristics and soil properties

(Gascuel-Odoux et al., 2011) and additional erosion and transport factors (Verstraeten

et al., 2000), exerting a more complicated influence on erosion control. For example,

the effect of landscape structure on soil erosion may be identified on moderate slopes,

while on steep slopes it may be concealed by on-site severe soil erosion (Chevigny et

al., 2014). However, the key process for erosion control is the fact that landscape

elements and their structural position (i.e. parcel structure, field boundaries, hedges



and similar) alter hydrological connectivity between land and water. This is particularly true when the land cover on both sides of boundaries is different (Van Oost et al., 2000). Reducing parcel size and heterogeneity increases hydrological

connectivity significantly and results in a substantial off-site damage effect, irrespective of on-site erosion of the investigated land use (Boardman et al., 2018; Deváty et al., 2019). During low and median flow conditions, surface runoff and sediment may arrive to a lesser extent at field boundaries due to efficient interception effects of the vegetation cover. This may explain the identified dynamic relevance of

land structure change in sediment load herein.

Our findings are also supported by the calculation of the management factor (C-Factor) and the slope and slope length factor (SL-Factor) of the RUSLE for Period I and Period II. While the mean C-Factor of the HOAL catchment increased from 0.16 during Period I to 0.33 for Period II, the SL factor increased from 0.76 to 0.96 from

Period I to Period I. Taken together, the changed values for these two factors increase the theoretical soil loss within the catchment by over 150%. This is smaller than the changes observed, however it should be noted that the RUSLE has not been designed to account for sediment loads of catchments but to estimate field scale soil loss within catchments. This may explain the observed differences to a certain extent.


## 5 Conclusions

Climate change, land use and cover change, and other human-associated activities are widely regarded as potential agents driving hydrological change. Understanding the



relevance of each of these factors in the hydrological cycle is critical for

implementing adaptive catchment management measures and addressing climate

change. For some hydrological cycle components, very significant climate change

influences in the last decades have been identified (e.g. Haslinger et al., 2019;

Duethmann and Blöschl, 2018). However, we found that climate change in rainfall

erosivity and precipitation cannot explain the increased sediment production between

1946-1954 and 2002-2017 in the investigated catchment. Instead, both land cover and

land consolidation played dynamic roles in controlling erosion.

Still, the relevance of land use and land cover change versus land consolidation

change varied dynamically with changing flow conditions. The reduction in parcel

density undoubtedly increases soil erosion risk, particularly at higher flows due to the

decreased capacity of trapping sediment particles between parcels and increasing flow

lengths inside parcels. Meanwhile, unfavorable land use or land cover change will

increase sediment load at most normal flow conditions, although the relevance of this

process would decrease at high or very high flow rates. Therefore, when addressing

soil conservation measures at the catchment scale, the distribution of fields, land

structure, and vegetation cover should be simultaneously considered. Such a strategy

would be conducive to dealing with the risk of soil erosions at different flow rates.

Land use policy adjustments resulting from technological development have been

vital to deal with food security issues in the past. However, now we have to

experience the negative influence of these adjustments on the hydrological cycle.

Therefore, rather than focusing on climate change solely, we need to pay increased



attention to anthropic management activities to counteract their negative impact on

hydrological change effectively.

**Author contributions**

Shengping Wang has led the data analysis, drafted the manuscript, and revised the
manuscript; Peter Strauss was responsible for the project design, oversaw the whole
analysis, and conduct manuscript revision as the project leader; Carmen Krammer was
responsible for data collection and data preparation; Elmar Schmaltz has contributed
to figure drawing and manuscript revision; Borbala Szeles has helped to revise the
manuscript, and Günter Blöschl oversaw and critically reflected on the manuscript
revision as the senior scientist.

**Competing interests**

The authors declare that they have no conflict of interest.


**Disclaimer**

Publisher's note: Copernicus Publications remains neutral with regard to jurisdictional
claims in published maps and institutional affiliations.


**Acknowledgements**

This work is financially supported by the SHUI project (Soil Hydrology research
platform underpinning innovation to manage water scarcity in European and Chinese
cropping systems) within the Horizon 2020 Research and Innovation Action of the
European Community (No. 773903), the Austrian Science Funds (FWF), project
W1219-N28, and the TU Wien Risk network.



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
