# Peer review of "Agricultural intensification vs climate change: What drives long-term changes of sediment load?"

_Hydrology and Earth System Sciences, 2021_

## Author Response (AR1)

**Authors response**
* * *
*The following documents the authors' response to reviewers' comments. Whilst the reviewers' comments are displayed in blue, the authors' comments/responses are displayed in black.*
* * *
Reviewer#1

I found the preprint titled "Agricultural intensification vs climate change: What drives long-term changes of sediment load?" an interesting and worthwhile contribution to the exploration and identification of potential drivers of long-term changes in suspended sediment. I applaud the authors on their well written abstract and introduction, use of data analysis techniques, and helpful interpretation of their results. I found much of the writing clear and concise. I have a few major comments, and some minor ones, that I hope the authors will consider.

RESPONSE: We thank the Reviewer for this positive evaluation. We revised the manuscript addressing the major and minor comments raised by the Reviewer as follows:

Major comments:

R1_1:Only 1 metric of "climate change" was evaluated in this paper and this metric, monthly mean precipitation, does not capture the change in climate most likely to effect sediment transport. Metrics such as change in the magnitude, frequency, or duration of the highest precipitation events (> 75th or 90th percentile? Max event?) would be more appropriate for assessing the effect of changes in climate on sediment transport. Furthermore, I don't see any discussion related to the type of precipitation (i.e., changes in the proportion of precipitation as snow versus rain). The catchment is located in Austria, so surely snow fall is a consideration. Is it possible there has been a shift from snow to rain which could be driving some of these changes in sediment?

RESPONSE: We agree that the metric of mean precipitation may not characterize erosion severity entirely. Therefore, we investigated both mean precipitation and erosivity density (Please see Figure 3 in the submission) which is calculated from peak event rainfall intensity ($I_{30}$) and kinetic energy of rainfall (E). This is a metric that has been developed within the frame of the most widely used USLE/RUSLE set of erosion models. Kinetic energy of rainfall and maximum intensity of rainfall commonly are both considered to be the main driving forces of rain on erosion.

Because we focus on the change in sediment loads, we think that such an index is a good indicator representing storm characteristics for erosion susceptibility.

Although the study area is in Austria, the HOAL is considered as a lowland catchment, it is located in a region with quite small amounts of snowfall. Snowfall in the catchment plays a minor role in sediment production, considering that melted snow may drive sediment transport for only a few events. So, in our analysis we think that considering the proportion of precipitation to snowfall is irrelevant when addressing the impact of climate change on sediment load. We included a comment to this effect in the revised paper (Please see lines 359-364 in the revised manuscript (clean version)).

R1_2: The differences in sampling methods between the two time periods needs more attention. As presented, I am not convinced we aren't just seeing changes in sediment due to changes in sampling methods (collection frequency and also techniques). Sediment is particularly sensitive to changes in sampling (see couple of references at bottom). Lines 228-230 suggest an approach for dealing with differences in sampling but it is not clear to me how this technique takes care of the potential issue. Even if the difference in sampling methods/techniques cannot be resolved, I think this analysis would still be worthwhile, as long as this issue was thoroughly discussed in the context of the results.

RESPONSE:

We fully agree that differences in sampling methods (in terms of sampling frequency and sampling technique) may affect estimated changes in sediment load in our analysis. As mentioned by the reviewer, addressing this issue is difficult with the available data. To address different sampling frequencies, we made use of the relationship of $a$ against $b$ which, to a certain degree, ensures that the values of the pairs of $a$ against $b$ lie nearly on the same straight line, as long as the sediment regimes represented by the pairs of $a$ and $b$ do not change significantly. In the revised manuscript, we have added more detailed explanations on the sampling methods used (Please see lines 247-260 in the revised manuscript (clean version)). Additionally, we added a section to discuss the possibility that different sampling techniques may have affected our observation in the discussion, and point out that less difference among various sampling techniques usually is found when the monitoring focuses on fine sediment textures such as in our study (Please see section 4.2 in the revised manuscript). Finally, looking at the total annual sediment loads that may be calculated only from sediment concentrations times water flow, it is clear that the total annual loads differ substantially between the two periods. Also, the comparison of measured data to some results of a quick erosion model application (Please see section 4.1) provides confidence in the results obtained. Thus, although some uncertainty exists

due to the different sampling strategies, we are confident that the different sediment loads are not only artefacts of the sampling.

R1_3:Statistical techniques can and should be used to help identify and quantify differences between periods. The text indicates a t test was used for assessing differences in monthly erosivity density (line 189) but how differences between the SRCs and the sediment loads were determined isn't described. I appreciate the development of the theoretical SRCs but even the differences between these are discussed qualitatively. For example, on lines 433-435, how is the distinction between "considerably higher" and "not different" being made? There is considerable overlap in the 5 and 95th percentiles in both panels of Figure 7. Specifically, on panel (a), the line for the 50th percentile for Period II falls outside the gray area for Period I. Using a statistical test like ANCOVA or a regression equation with categorical variables for season and time would help to determine which slopes and intercepts are (statistically) significantly different.

RESPONSE: We have revised our manuscript by using ANCOVA for the comparison of SRCS, the relationship of coefficient a against b, and the derived tSRCS between seasons and/or periods. In the revised manuscript, we explicitly point out, that the fitted SRCs (Figure 5 in the revised manuscript) were significantly different between growing seasons or dormant seasons.

Whilst for the relationship of coefficient *a* against *b* (Figure 6 in the revised manuscript), most parameters of these relations (in terms of the slopes and intercepts) were not significantly different at *p* level of 95%, except for the intercepts in the growing season. Thus, in the revised manuscript, we have rephrased the comparison of the relationships of coefficient a against b (Please see lines 470-474 in the revised manuscript) (clean version).

As for the tSRCs, we also found that the median tSRCs were significantly (*p*<0.05) different between the growing seasons or dormant seasons. However, to account for uncertainty of the tSRCS, we have rephrased the comparison of tSRCs (please see lines 491-503 in the revised manuscript) (clean version).

As for sediment loads, we used t-tests to examine the difference, and we have added this statement in the revised manuscript.

R1_4: Given the structure of the dataset, how were the sediment loads presented in this report calculated? What technique was used? Relatedly, a table of these loads might be worth including since the term "load" is featured in the title. Also, Lines 459-460 used the word "load" to reference to Figure 8 but this figure is showing sediment concentration. In several places in the paper it feels like the terms "concentration" and "load" are being used interchangeably. Sometimes this can be ok since the SRCs are always positive and increases in concentration can be inferred to

mean increases in load, but they are not the same thing and it would be prudent to be clear about which term is used when presenting the results of these analyses (i.e., these SRC are build using sediment concentrations not loads).

RESPONSE: We have added a short paragraph to explain how we estimated the statistic of sediment load (Please see lines 201-213) (clean version). We also present an additional table in the revised manuscript as well to show the statistical results (See Table 4). It is true that we have not sufficiently addressed the terms load and concentration and have occasionally mixed them. Thank you for pointing this out. Thus, we have revised the word "load" to "concentration" when referring to Figure 8. We also have checked the whole text to make sure the word "load" and "concentration" not misused consistently.

R1_5: I'm wondering about the choice to average concentration and streamflow data by month…. I wonder if some of the important variability (related to the magnitude of the events) is being muted? The report states that a majority of the sediment load is transported in just a few high flow events in each period, but these important events are being averaged with all the available data for each month.

RESPONSE: Yes, we also realized that the use of monthly average data may suppress some temporal variability of events. It is the discrepancy in sampling frequency that made us use monthly sediment concentrations, as we would like to understand the change in sediment regime due to land surface process without the potential interferences of sampling methods.

We agree that a monthly dataset may be not useful for understanding event processes, however, it is at least helpful for understanding seasonal or annual scale alterations of the transport regime between different periods. Because peak values of a sediment graph or hydrograph usually do not last for a long time in our catchment (size 66 ha), we think that it is less important to account for peak observations when calculating or estimating sediment load by month. It has to be noted though, that we have occasionally made improper use of the term "event" when talking about sediment loads because all our considerations refer to monthly loads. Thus, we have checked through the whole text for ensuring proper using the terms.

Minor comments:

R1_6: Line 245: Is this saying the solid and dashed lines in 2b represent a shift in the sediment transport regime such that concentrations at a given discharge are relatively lower for periods A', B', and C' compared to A, B, and C? Consider rewording.

RESPONSE: Yes, this means a sediment regime shift represented by the two lines. We have rephrased it as "As for different lines in Figure 2b, the lower ones characterized by points A', B', and C' represent situations with most of the annual sediment load being transported at relatively low flow discharges, whilst the higher ones characterized by A, B, and C represent situations with suspended sediment being mainly transported at high streamflow." (Please see lines 270-274) (clean version).

R1_7: Line 226: The text indicates SRCs were also fit by season-and-year however the results and discussion only present and discuss the SRCs by season. Suggest removing this since the by year and season results aren't discussed.

RESPONSE: Actually, we fitted the regression by each specific year's season (for example, growing season of 1950, 1951,...), instead of a pooled season (for example, growing season or dormant season). In the revised manuscript, we have rephrased this as "fitted by each specific year's season".

R1_8: Line 269: Figure 4 is used to support the statement that the coefficient of log a follows a normal distribution, but Figure 4 is a flow duration curve. Also "shown" would probably be a better word than "proven" in this context.

RESPONSE: Thank you for pointing this out. We have corrected this in the revised manuscript and replace the word "proven" with "examined".

R1_9: Figure 3: Consider including labels that indicate which months are in the Growing vs. Dormant seasons. Are the months included in the growing and dormant seasons available somewhere in the text? I can't seem to locate it. My apologies if I just missed.

RESPONSE: Thanks for the suggestion. Growing season includes the months from April to October, while dormant season includes the months from November to March. In the revised manuscript, we have added a short explanation of this in the caption of Figure 3 "Figure 3 Monthly erosivity density a) and monthly precipitation b) for Periods I and II. The bars with a dashed outline represent the growing seasons (April to October), the bars with a solid outline the dormant seasons (November to March). The whiskers indicate the range between the minium and the maximum, and the astersks the outliers". We have also shown this information in the methods section (Please see line 188 in the revised manuscript) (clean version).

R1_10: Figure 1: Caption indicates a "black hatched area in b", but I don't see it. How different are the catchment sizes between the 2 time periods?

RESPONSE: The difference in size of catchment between the two periods is very small (around 200 m$^2$), which is barely visible on the map, a small area only around the catchment outlet. We have revised the figure caption, and also replaced this text

with an explanation on the change of catchment sizes in the Methods section (Please see line 143-144) (clean version).

R1_11: Line 330: Can you support this statement more? I think the authors are saying that decreases in streamflow cannot account for the observed increase in sediment transport, because if that was the case, then we'd expected to see increased streamflow, correct? Consider rephrasing

RESPONSE: In the revised manuscript, we have rephrased this as "The decreased flow regime of Period II, probably partially due to increased evapotranspiration in the last decades (Duethmann and Blöschl, 2018), indicates that streamflow cannot account for the increased sediment regime over Period II, because if that was the case, we would expect increased streamflow in Period II."(lines 370-373, clean version). Please also note, that for a given catchment, there is usually a positive relationship between flow rate (be it annualy or monthly) and sediment load. Thus, if the flow has even decreased, this clearly points to a nonsignificant influence of flow rates on increasing sediment loads.

R1_12: Figure 6: Keep the colors used for Period I and II consistent between Figures 5 & 6. What are the arrows for? The text describes the right points being the "left-upper area" but this is not true for (a).

RESPONSE: We have changed the colors of Figure 5 to be consistent with Figure 6 . We have also explained the meaning of the arrow in the caption of Figure 6 "The arrows represent sediment regime shifted upward to a certain degree". As for the point distributions of Figure 6, we have rephrased the expression for clarity to "Interestingly, the values of both Periods I and II, in the growing season more concentrated in the right-lower area (Figure 6a). A different pattern of log (a) against b was found for the dormant season (Figure 6b), i.e. the data points of Period I concentrated in the right-lower area (blue points), but were more concentrated in the left-upper area for Period II." (Please see lines 459-463 in the revised manuscript) (clean version).

R1_13: Lines 436-440: The +/- ranges for the loads given in this paragraph would result in negative sediment loads. Also, +/- ranges are quite large suggesting perhaps there isn't a statistical significant difference between these loads?

RESPONSE: Unfortunately, annual sediment loads of small catchments usually have a large heterogeneity. Therefore, different periods of observation may exhibit large differences for annual loads such as in our case. To account for this statistically, in the revised manuscript, we rephrased the paragraph on loads. Specifically, we examined statistical difference between the loads by using a t-test. Also, we examined the statistical difference of the derived tSRCs by using an ANCOVA. We found, that sediment loads in the growing season were significantly different between Periods I and II, but they were not significantly different for the dormant season. As for the

tSRCS, we found that $tSRC_{50\%}$ were significantly different between two growing seasons or dormant seasons. In the manuscript we have emphasized that, when accounting for uncertainties of tSRCs, their difference varies with flow rates: "ANCOVA suggests that the derived tSRC50% were significantly different ($p<0.05$) between the two periods, both in the growing seasons and dormant seasons. However, when accounting for uncertainties of the derived tSRCs, the degree of their difference varied with flow rate. Specifically, at a given Q higher than approximately Q70%, sediment concentrations in Period II_G were higher than those in Period I_G (Figure 7a), whereas for flow rates below this value concentrations were not different. " (Pleases see lines 490-496 in the revised manuscript) (clean version).

R1_14:   Figure 8: What does the dashed vertical line at zero represent?

RESPONSE: In fact, when zooming into Figure 8 to check the variation of either parcel_effect or LUCC_effect at low flow conditions, we found that their values exhibited little plausible dynamics below a flow rate of 2 l/s. We mainly attributed this behavior to an artefact of calculation. In fact, sediment concentrations at low flows (no events) are almost zero in both periods, because sediment transport largely depends on events. Therefore, in our calculations, we focused on the sediment – flow relationships for flow rates above a base flow of 2 l/s. 'The dashed vertical line that is visible near the zero value (in fact it is the value of 2 l/s) is an artefact of this calculation procedure. In the revised manuscript we have changed the minimum value of the x-axis to 2 l/s to avoid further misunderstanding of this figure.

R1_15:   References on sediment sampling in rivers:

Awal, R., et al., 2019, A General Review on Methods of Sediment Sampling and Mineral Content Analysis, Journal of Physics: Conference Series, https://doi.org/10.1088/1742-6596/1266/1/012005

Groten, J.T., and Johnson, G.D., 2018, Comparability of river suspended-sediment sampling and laboratory analysis methods: U.S. Geological Survey Scientific Investigations Report 2018–5023, 23 p., https://doi.org/10.3133/sir20185023

Harmel, R.D., et al., 2010, Impact of Sampling Techniques on Measured Stormwater Quality Data for Small Streams, Journal of Environmental Quality 39:1734–1742, doi:10.2134/jeq2009.0498

RESPONSE: Thanks for the valuable reference information. We have added the relevant references to the revised manuscript.

■■■■■■■■■■■■■■■■■■■■■■■■■■■■■■■■■■■■■■■■■■■■■■■■■■■■■■■■■■■■

■■■■■■■■■■■■■■■■■■■■■■■■■■■■■■■■■■■■■■■■■■■■■■■■■■■■■■■■■■■■

**Authors response**
* * *
*The following documents the authors' response to reviewer' comments. Whilst the reviewers' comments are displayed in blue, the authors' comments/responses are displayed in black.*
* * *
Reviewer#2:

General comment

R2_1: Language not well written for the most part but with well written sections. Lots of od sentences. Lots of missing "the"'s (I am not a native English speaker).

RESPONSE: We have edited our manuscript, by revising the language.

R2_2:The structure of the paper is good. But with some short comings. The introduction needs to be more informative and more relevant to the rest of the paper. The is some information lacking about the sediment and water flow data.

RESPONSE: We have revised the introduction to account for this comment (R2_2).

R2_3: The paper spans 72 years of data but only actually utilizes 25 of these, which is a pity. I think the paper would be stronger with for example 6, 12 year periods instead of 9 year in the beginning and 16 in the end.

RESPONSE: The reason of this choice was that the data set of discharge and sediment concentration for Period I was only available for the time between 1945-1954. After that, measurements were stopped and only started again in 1990. We used the wording of "72 years" to emphasize the relevance for climatic change analysis. However, although only a limited amount of data is available for period I, we think that their information content is extremely valuable because to our best knowledge, for this period of time (1945-1954), almost no sediment concentration data are available. We have addressed this issues in the revised manuscript (Please see lines 136-142) (clean version).

R2_4: Line 27-28,   it is a 36% increase, i would say that, that is a significant increase.

RESPONSE: Thank you for pointing this out. We have checked the statistics and agree. We have rephrased the text to "given that the mean daily streamflow significantly decreased from $5.0 \pm 14.5$ $ls^{-1}$ for Period I to $3.8 \pm 6.6$ $ls^{-1}$ for Period II."

R2_5: Line30-31, have these been shifted? Wont this give the highest yields in period 1? Number of decimals is probably too high.

RESPONSE: Thank you for pointing this out. We have updated the slope values and reduced the number of decimal places.

R2_6:   Line36,    medium?. <Q20% is low not medium. (or state that Q20% is the highest 20%?)

RESPONSE: We consider "Q20%" as the highest 20%. This means, for Q less than Q20% approximately, both median and low flows are included. To make this sentence clearer we have changed it to: "At low and median streamflow conditions, land consolidation in Period II (i.e. the parcel effect) had no apparent influence on sediment production".

R2_7: Line 39, increasingly important?

RESPONSE: We have rephrased this part to "parcel structure became more important in controlling sediment yield".

R2_8: Line 36-41, consider rephrasing

RESPONSE: We have rephrased this part to "With increasing stream flow, parcel structure became more important in controlling sediment yield, as a result of an enhanced sediment connectivity in the landscape, leading to a dominant role at high flow conditions." (Please see lines 38-41 in the revised manuscript) (clean version).

R2_9: Line 40-41,    is 7day/year = extremely high, I think it is just high.

RESPONSE: We agree. Terminologies such as "extremely", or "significant" should have a conceptual basis when used. We therefore have deleted "extremely" and rephrased this sentence to read "With increasing stream flow, parcel structure became more important in controlling sediment yield, as a result of an enhanced sediment connectivity in the landscape, leading to a dominant role at high flow conditions."

R2_10: Line 41 , in period 2 or between period 1 and 2?

RESPONSE: We have revised this sentence to "between Periods I and II".

R2_11: Line 41-43,  rephrase, consider deleting the last part. Unfavorable? = increased erosion => increased suspended sediment transport?

RESPONSE: Yes, "Unfavorable effect" herein means an increased sediment regime. To make this clearer, we revised this sentence to "The increase in cropland between Periods I and II at the expense of grassland led to an increase in sediment flux, although its relevance was surpassed by the effect of parcel changes at high flow conditions".

R2_12: Line 44, explain "land consolidation" and "parcel structure"

RESPONSE: We use the terms "land consolidation" and "parcel structure change" almost interchangeably. The term "land consolidation" is intended to mean parcel structure change due to land use policy adjustment which aims at enhancing agricultural intensification by combining scattered land parcels into bigger ones. This is a land policy terminology. We suggest keeping this terminology because it is of common use and directs the reader towards the bigger picture of policy driven land structure changes related to agricultural industrialization.

R2_13: Line 54, soil erosion is a process or a phenomenon not a risk.

RESPONSE: We agree and replaced the term with "a phenomenon".

R2_14: Line 61-63,    do you mean the effect of LULC change on soil erosion?

RESPONSE: Yes, it means the effect of LUCC on soil erosion. We added the abbreviation LUCC to the text to be used later on.

R2_15: Line 63-65,    what did they find?

RESPONSE: Our research purpose was to understand the respective effects of climate change, LUCC, and landscape structure change. A lot of studies have been previously carried out on individual effects. For those studies paying only attention to revealing the impact of LUCC, we do not want to use more space to introduce their findings. We therefore simply mentioned that this kind of analysis has been conducted previously.

R2_16: Line 66,  consider using LULC

RESPONSE: As a consequence of introducing LUCC in line 60, we can use the abbreviation now - thanks for the suggestion.

R2_17: Line 68, References needed

RESPONSE: Thank you for pointing this out, we have added additional references.

R2_18: Line 71, Wet or moist? A climatic period is usually considered to be 30 years, I guess that this refers to a shorter period, consider using "weather" instead.

RESPONSE: In the revision of the introduction (as requested in comment R2_2) we have deleted this part.

R2_19: Line 71-73, rephrase. Does this mean that increased sediment loads were only found as a result of prolonged/more severe drought periods? Is this in contrast to earlier periods? And please explain why.

RESPONSE: In the revision of the introduction (as requested in comment R2_2) we have deleted this part.

R2_20: Line 73-77 In what direction was the contributions? Consider rephrasing

RESPONSE: We have rephrased the latter part of the sentence to "with the contributions being +29%, +40%, and +31%".

R2_21: Line 75, sediment reduction? Do you mean reduction in sediment load?

RESPONSE: Yes, it means reduction in sediment load. We have rephrased this to "reduction in sediment load".

R2_22: Line 78, and engineering measures

RESPONSE: Thank you for pointing this out. In the revision of the introduction (as requested in comment R2_2) we have deleted this part.

R2_23: Line 78-79, during "the period" instead of over?

RESPONSE: We prefer "over" to "during" here, because we would like to emphasize the trend of the reduction.

R2_24: Line 84-86, In what way? Make relevant.

RESPONSE: We will rephrase the sentence to "The previous findings provide valuable information on understanding how land use and/or climate change affect soil erosion and/or sediment load". (Please see lines 74-75 in the revised manuscript) (clean version).

R2_25: Line 86-87, please explain "land consolidation", "landscape structure" and what it has to do with land use polices.

RESPONSE: The term "land consolidation" in our study means combining small land parcels into bigger ones. This usually causes landscape structure to change because field edge structures such as hedges disappear. In our analysis both terms refer to the same human activity, and sometimes we use them interchangeably. The term "land consolidation" has also been widely used, as a plan or policy in many countries, such as Austria (since 1955), the Czech Republic or China. There exist even national agencies to carry out land consolidation plans. The main reasons behind the policy of land consolidation are to enhance agricultural intensification, to be conductive to using advanced machine technology, and thus to be helpful for easier land management.

R2_26: Line 84-91, consider splitting sentence

RESPONSE: Thank you for the suggestion, we have splited this sentence into more. (Please see lines 76-83 in the revised manuscript) (clean version).

R2_27: Line 93, land uses and land units? What does this mean? Is it that field edges/margins usually has permanent vegetation and therefore some trapping capacity, and therefore smaller fields will have smaller soil losses than larger fields because they have more field edges?

RESPONSE: To our understanding, boundary effects do not necessarily related to permanent vegetation on edges/margins, although this is often the case. Boundary effects may also result from different field properties (such as field infiltration, water/sediment trapping capacity, and so on) between two adjacent land parcels. This may for instance be the case when two fields are managed in a different way, say one is without cover, another one has some soil cover developed. For a given crop, a small field will exhibit smaller infield (onsite) soil losses compared to a larger field, but this is due to the effect of the field properties which is different from processes that occur at the edges of fields.

R2_28: Line 100, studies not authors (they are "et al.")

RESPONSE: We have revised this line accordingly.

R2_29: Line 92-101, what did they find beside that parcel structure matters. Are small better than large, long better than round…? You should enlighten the reader.

RESPONSE: The key findings of both studies are to underline that inner organization of field blocks have a strong effect on the risk of soil erosion, and both studies are dedicated to revealing the relevance of boundary effects on soil erosion controls from the perspective of hydrological/sediment connectivity. Both studies did not explicitly conclude that "small blocks are better than large" or "long better than round". To our understanding, hydrological/sediment connectivity is essentially resulting from landscape heterogeneity, instead of a simple size effect or shape effect. Thus, after

consideration we suggest to show only the actual findings of both studies and keep the text as is.

R2_30: Line 101-105, what did they find? Anything the reader should know about, otherwise consider deleting

RESPONSE: This part is intended as a short summary for all of the previously mentioned studies. Since most of their findings have been illustrated in the previous paragraphs, we did not want to expand this section to report their results/conclusions more than necessary. However, as a summary we still think that these lines are helpful and thus suggest keeping them unchanged.

R2_31: Line 108 , <suggestion> showing that the same land structural changes have different impact in different landscape and agricultural settings?

RESPONSE: Yes. According to studies such as David et al. (2014) and Cantreul et al. (2020), the impact of land structure change on soil erosion would be different in different environmental settings, in which underlying soil and/or vegetation are different.

R2_32: Line 114, what practical perspective?

RESPONSE: During the revision process of the introduction, this sentence has been deleted.

R2_33: Line 115 is this the first time land use and land cover change is mentioned

Response: We have revised and defined the term "LUCC" at the place where it was mentioned first (line 60 in the revised manuscript) (clean version).

R2_34: Line 121, parcel structure change? Is the change made with the intention of reducing erosion?

RESPONSE: The change has been made without any consideration of the soil erosion problem. It is a consequence of agricultural industrialization that has taken place since the 1940's. Like with many changes during this period of agricultural industrialization it now turns out that very negative side effects exist (in our case erosion). To better specify this, we have revised the sentence to "has experienced a significant change in land use and land cover as well as parcel structure due to altered land management policies during the past decades".

R2_35: Line 122, sediment concentration? And yield/load calculated from these? Is it suspended sediment or total load, including bed load?

RESPONSE: We have revised this to "suspended sediment concentration". Sediment yield/load is calculated from these observations. There is no bedload existing in the

HOAL stream, because it is located in an area which consists of fine tertiary sediments and the soils do not have any stone content.

R2_36: Line 126,  I see that the two ends of the period are interesting to compare, but the inter long period in between is also of interest and could/should act as kind of validation period. What if you find differences between the end periods, and draw conclusions on these but these conclusions might not explain what happened between 1954 and 2002. Why 9 years between 1946 and 1954 and 16 years between 2002 and 2017. I guess 9 years is a relatively short period could be influenced by "extreme years".

RESPONSE: We completely agree that a long-term observation records is valuable and important for understanding the trend of the change in sediment load/yield. However, the datasets of sediment concentrations and flow rates between 1954 until the 1990's are not available, due to a lack of measurements. Therefore, we are compelled to restrict our investigations to the described time periods. What we do have, is a record of climatic observations over 1946 to 2017. On the one hand, according to the long-term observations of precipitation, we found that there are no significant trends (p>0.05, Mann-Kendall test) with respect to climatic parameters over the past decades (see the following Figures). This reinforces our confidence in comparing the two (relatively) short-time periods. Additionally, the focus of our study is to compare the SHIFT and/or change of sediment regime between the two periods, rather than a TREND of sediment regime over 1945 to 2017.

[Figure]

Figure1 Time series of climatic parameters for the HOAL Petzenkirchen; left upper: average annual precipitation, right upper: number of erosive events, left lower: average rainfall erosivity as defined in the RUSLE soil erosion model; right lower:

average erosion density (this was used in this study as a climatic indicator for the change in erosive potential rainfall) The red line represents the 15 years moving averages.

R2_37: Line 55-114   The introduction mentions many relevant subjects and lots of references but does not really make them relevant for the study. The reader does not get much better prepared for reading the paper, by reading the paper. Consider focusing on fewer and the most relevant subjects and let the reader know what all/some of the studies referred to found and why this is relevant to the present study/paper.

RESPONSE: We have reorganized the introduction to account for this comment. Specifically, we confined relevant studies to a few of attribution analysis. We then point out that most of the previous analyses considered LUCC and landscape structure change as a whole to understand it's role in affecting sediment load. Next, we emphasized that the relevance of landscape structure changes has so far received less attention, even though land-use policies, such as land consolidation, have been changing agricultural practices. We now also elaborate on the relevant studies about the impacts of landscape structure change subsequently, which is then followed by our research purposes. R#1 emphasized the quality of the introduction, and we therefore kept the changes to a moderate level.

R2_38: Line 133        is there a map somewhere?

RESPONSE: Yes, there is Figure 1. R2_44 also addresses this point.

R2_39: Line 146        suspended sediment (or suspended matter, thus including the organic part). Please be more precise here.

RESPONSE: Thank you for pointing this out. We have revised this to "suspended sediment concentration".

R2_40, Line 150        "at" not "by"? and using some water level/stage measuring devise and calculating flow using a stage-flow relationship (Q/h)?

RESPONSE: We prefer "by" over "at" here, because we want to emphasize the observational approach, i.e. "a Thompson weir and a paper chart recorder". Streamflow was calculated by water level and a flow rating curve (i.e. a stage-flow relationship).

R2_41: Line 152     measured "manually" meaning that samples were collect manually? Through bottle dipping?, surface/depth integrated?

RESPONSE: Yes, this was done by bottle dipping directly into the mixing zone of the weir. The stream is very small (baseflow of about 2 l/s), so mixing is reached quickly. There is therefore no need of depth integration unlike in larger rivers.

R2_42: Line 153    automatic method, meaning? ISCO samplers/turbiditymeter? What is the additional manual sampling used for and how often is it collected, and what is the time resolution of the unspecified automatic method?

RESPONSE: In the second period we used 2 ISCO samplers (48 bottles). The start of the sampling was triggered by increased flow rates to obtain samples for the rising and the falling limb of the hydrograph. The sampling interval for the first sampler was set to 15 min, the second sampler was set to 1 h. In addition, manual sampling took place once a week in Period II.

R2_43: Line 156        what is "the vegetation period"?

RESPONSE: We have changed the text to "growing season".

R2_44: Fig 1    a) Ok to zoom in a bit. b & c) the gauge looks like it is in the same place in both maps. I guess it should be moved down stream for 1946? In the text "paved" or "roads and settlements (line 159)"is used rather than "sealed", please chose one and correct throughout. What are the lines within the catchment, roads? I don't think they are part of the "symbol key" or "legend"

RESPONSE: The gauges in Fig b) and c) are located at different places. This makes a small, negligible difference in catchment size (about 200 m$^2$). We have cleirified this point in the text (Please see Lines 142-144) (clean version). As for the "roads and settlements" , we rephrased this to "roads and settlements (i.e. paved area)", and "paved area" was used throughout the following text. The lines within the catchment refer to the stream or to contours. This is indicated in the legend for this Figure. Please see also #R1_10.

R2_45: Line 175        This is the kinetic (potential erosive power) of rain events. You cannot say what the effect of rainfall is quantified by this measure as it also depends on the erodibility.

RESPONSE: We agree that soil erosion is dependent on many factors such as rainfall erosivity, soil erodibility, and some others (such as crop management, conservation practices, and topography). However, in this analysis the focus is on how climate change\weather change affect soil erosion. The parameter EI30 is a combination of kinetic energy and maximum intensity of rain. It is a commonly accepted parameter to describe the erosive power of rain. Our intention in using this parameter is to have an indicator for the power of rainfall to cause erosion, because one of our assumptions at the beginning of our study was that the erosive power of rainfall has increased due to climate change. We could show that this was not the case for the studied periods. Soil

erodibility (however is defined and measured) is a measure of the susceptibility of the soil against erosion and does not affect EI30. We agree that a change in soil erodibility may have influenced the erosion susceptibility of the catchment between the two periods. The main driving force of soil erodibility is soil texture which has not changed during the periods. Furthermore, organic carbon of a soil is usually a parameter that may modify soil erodibility, but to a much lesser extent compared to soil texture. Only few results on soil organic carbon are available from Period I, but they suggest no substantial shift in organic carbon contents between the two periods. Therefore, we are confident that a change in soil erodibility is not a major reason for the differences in sediment loads.

R2_46: Line 184      event precipitation?

RESPONSE: Yes, we calculated rainfall erosivity for each precipitation event according to 5-min rainfall intensity data records, and then estimated the statistic of rainfall erosivity by month and by year, respectively.

R2_47: Line 184-186      please explain why

RESPONSE: We have added an explanation to the revised manuscript "We assumed erosivity density ED (i.e. EI30 divided by event precipitation) to be a particularly relevant climatic indicator of the soil erosion process and catchment sediment yield because it is calculated as a combination of rainfall kinetic energy and maximum rainfall intensity of rain events. These are commonly considered as the relevant parameters of rain to trigger the soil erosion process." (Please see lines 181-185 in the revised manuscript) (clean version). R (or EI30) is commonly a measure of the erosion force of specific rainfall, determined by the amount of rainfall and rainfall peak intensity. ED is calculated as EI30 divided by event precipitation, so ED actually addresses the impact of the difference in the amount of precipitation. Thus, ED is an indicator of erosional forces of rainfall irrespective of different amounts of precipitation.

R2_48: Line 189      but wouldn't you expect that this is the period where it is most relevant? As there are no/less crops/vegetation in the fields.

RESPONSE: Less vegetation is of course present during the dormant season. However, the main driving force of erosion remains erosive rainfall. Erosive rain may very well be characterized by EI30. Erosive rainfall does not take place during winter in Austria, therefore the risk of soil loss during the dormant season is very low compared to the vegetation period. We may refer to Figure 3a which demonstrates the monthly distribution of erosive rainfall at the site (very similar to the rest of Austria and Central Europe by the way) to show, that the risk of soil loss during the winter months is very low compared to the soil loss risk during the vegetation period. This is also the reason why we separated our analysis into dormant and vegetation period.

**R2_49: Line 198-192** how good/bad is this relationship (R2=0.xx)? Is it validated against other periods?

RESPONSE: This relationship was established by using the dataset of 2006 to 2017, and it performed well with $R^2=0.82$ (n= 12) (Please see the following figure). Also, it performed well for the validation period of 2000 to 2005, with $R^2$ over 0.8.

[Figure]

Figure 2 Correlation between EI30 and Rain for Period II (2006-2017)

**R2_50: Line 192** show some number that indicates that the erosivity is low, in contrast to some higher numbers.

RESPONSE: Thank you for pointing this out. We have revised this as "Erosivity density is very low during the dormant season (the mean $ED$ was $0.66\pm0.21$ and $2.54$ $\pm2.43$ MJ ha$^{-1}$ hr$^{-1}$ respectively for the dormant season and growing seasons of Period I, whilst it was $0.42\pm0.11$ and $1.87\pm1.35$ MJ ha$^{-1}$ hr$^{-1}$ respectively in Period II)" (Please see lines 191-195 in the revised manuscript) (clean version).

**R2_51: Line 195** the transport capacity of suspended sediment is usually not used to its full limits (the stream/river is capable of transporting much more than it actually transports), due to a limited supply of suspendable matter.

RESPONSE: We agree but suggest keeping the sentence as is, because we only intended pointing to the possibility that transport conditions may have changed.

**R2_52: Line 201** assumed? Didn't you test this?

RESPONSE: Yes, we tested this. Here we refer to choosing a possible function, following the commonly methods applied in the relevant studies, before we fitted the model.

R2_53: Line 215 delete "the other studies". But did these studies also have data on a much better resolution than months? I think you are missing a chance to see shorter term sediment dynamics here. Don't you have data on 3-4 days intervals in period 1 and even better (but still unspecified! (I see now)) for period 2? Why not utilize this better resolution. (by reading on I see why a monthly resolution is chosen. But here it sounds like it is something you need to do because of the different sampling resolution. Consider just stating that the further analysis does not require the individual samples but an aggregate of these. Still it might be possible to use a smaller time resolution.

RESPONSE: Thank you for the comments and suggestions. We also agree that a finer sampling resolution often is better than a coarse resolution for understanding key processes and mechanisms. However, the sampling resolution is different between the two time periods. Usually this complicates the detection of changes in the sediment regime due to other processes (such as LUCC, landscape structure change, and climate change). To minimize the potential effect of different sampling resolutions, we used an aggregated monthly dataset as a possible way to address this issue. Additionally, we employed the relationship of slope a against b. Thus, we have used two different approaches to address the issues of different sampling resolutions. Similar questions were also raised by Reviewer#1. For additional details, we refer to the response to #R1_5.

R2_54: Line 216-218 but you will also have months that are dry in one end and wet in the other end, how will the approach work in such a case. And cases where the precipitation is normal in a month, but the past two months have been either very wet or dry, which will give very different responses in flow and maybe in sediment load in the month in focus.

RESPONSE: We understand that antecedent precipitation might have an influence on sediment supply and sediment load subsequently. However, we think that, at the scale of monthly variation, this effect is less important compared to the event scale. For a individual monthly analysis, it might be worth accounting for the effect of antecedent precipitation, but for the analysis with an intention of fitting a regression on a long-term dataset, it is possible to understand the dominant controls of monthly precipitation in sediment load by directly relating monthly P and sediment concentration.

R2_55: Line 225 does a power function has "a slope"?

RESPONSE: When using the power function, the common practice is to take log-transformed values for both sides of the equation. In the log-space, the coefficient "a" is an intercept, and the coefficient "b" is a slope, and in the reference literature, the coefficient "b" is commonly termed as a "slope".

R2_56: Line 229        do you expect that the different sampling strategies and sampling equipment bias the measured concentrations? Wouldn't it "identify" more than "exclude"?

RESPONSE: Yes, different sampling strategies were commonly found to influence measured concentrations to a certain degree (e.g., Thomas, 1988; Groten and Johnson, 2019; Harmel et al., 2010 (For details, please refer to the reference list in the revised manuscript)). To overcome the disadvantage of different sampling approaches, we tried to make use of the relationship of a against b. By using the framework of *a* against *b*, it is possible to examine the shift of sediment regime that resulted from land surface processes changes, concerning the interference of different sampling approaches. The original sentence (i.e. "this technique could exclude the interference of different sampling practices with the estimated sediment regime") has been deleted in the revised manuscript. Instead, we have reorganized the description of the framework of coefficient a against b and add additional text. (Please see lines 243-260 and section 4.2 in the revised manuscript) (clean version)

R2_57: Line 231-264        consider if this, or all of this, is necessary.

RESPONSE: Thanks for the suggestion. However, this section is important to understand the methods and analysis throughout the whole manuscript. Therefore, we suggest keeping this section and providing additional explanations on the method for answering the comment and/or question of R1_2.

R2_58: Line 283    why is landscape structure affecting the runoff in the dormant period? The evaporation won't be much affected? And is low anyway.

RESPONSE: In order to address this comment, we have revised this to "landscape structure in the dormant season was considered a determinant factor for water and sediment transfer across the land surface, and thus runoff and sediment production".(Please see lines 310-312 in the revised manuscript) (clean version).

R2_59: Line 287        it is therefore also assumed that the majority of the sediment in the stream origins from the fields (or the landscape) in contrast to originating from bank erosion and drainpipes etc. It would therefore be good if you could provide a sediment source apportionment, if possible.

RESPONSE: Thanks for the suggestion. The area of our catchment is only 0.67 km$^2$. No obvious change was found in the shape of the small stream for the two periods, nor did stream incision take place based on comparison of aerial photographs. This suggests that the main contribution of sediment is from the hillslopes. We consider the transport of sediment as a stepwise process. During erosion, sediment is detached from the slopes and transported towards the open water. Within one event, it may or

may not be transported directly into the brook (depending on the magnitude of the event). It is very probable that sediment detached at one place in the catchment will be deposited at another place and will finally reach the brook sometimes after several events. Detailed knowledge about this process unfortunately is not available yet. Eder et al. (2014) did a study on resuspension of the sediment within the HOAL stream. They found sediment concentrations due to resuspension to be quite low. We have included this reference to the revised manuscript and added this information to the manuscript. (Please see lines 315-318 in the revised manuscript) (clean version).

Eder, A., Exner-Kittridge, M., Strauss, P., Blöschl, G., 2014. Re-suspension of bed sediment in a small stream – results from two flushing experiments. Hydrol. Earth Syst. Sci. 18, 1043–1052.

R2_60: Line 198-302    well written section. Having read this I think that it is a pity that you don't use the whole data set, 1946-2018ish. You claim in the abstract that you span 72 years but actually only about a third is used. Why not split the whole period into for example 6, 12-year periods? It would make the analysis more robust and interesting.

RESPONSE: We completely agree that it would be better to use the whole time period of 72 for a complete analysis. Unfortunately, the data collection of flow and sediment concentrations was suspended after 1954 and only started in 1990 again. We also may refer to our comments on the remarks #R2_36 which point into the same direction.

R2_61: Line 305    "climate change" or "variations in weather"

RESPONSE: We have changed this to "climate change".

R2_62: Line 306-308    how do you find out if climate (variation/change) affects the variation of/in sediment regime by comparing erosive density and monthly precipitation?

RESPONSE: The term "climate change" may include a lot of indicators to describe its effect on a variety of parameters. We were interested in the potential effect of climate change on water erosion. Therefore, we were looking for climatic indicators that would allow us to analyze a potential change in climatic parameters that would have an effect on sediment load. The indicator erosivity density is composed of a calculation of EI30 divided by rainfall amount. The EI30 parameter is the most commonly used parameter to describe erosive power of rainfall (see for instance: Mark A. Nearing, Shui-qing Yin, Pasquale Borrelli, Viktor O. Polyakov, Rainfall erosivity: An historical review, CATENA, Volume 157, 2017, Pages 357-362, https://doi.org/10.1016/j.catena.2017.06.004.) Thus, we are confident that a change of

erosivity density would be a very good indicator for identifying a potential climatic effect. We also think that the amount of precipitation is an excellent indicator for identifying climatic effects on soil loss, because for otherwise constant conditions of management, total runoff in a catchment is quite strongly related to total precipitation and total sediment load.

On the other hand, the main purpose of our analysis is to disentangle the impacts of LUCC, land structure change, and climate change. Whilst a lot of studies have used various methods to separate the impact of climate change and LUCC change, few have disentangled the impacts of LUCC and land structure change. We therefore paid particular attention to the impacts of LUCC and landscape structure change, whilst for the impacts of climate change, we used less space to qualitatively estimate it's impact according to the change in P and erosive density, instead of quantitatively estimating it's contribution by such as modeling approaches.

R2_63: Line 309-310    I guess that 2,4 is period 1 and 1,8 is period 2, but it is not mentioned

RESPONSE: Thank you for pointing this out. We have rephrased this to "The mean monthly ED in the growing season were $2.37 \pm 1.38$ and $1.84 \pm 0.86$ MJ.ha$^{-1}$.hr$^{-1}$ for Periods I and II, respectively, but no significant difference (p>0.05) was found between the two periods". (Please see lines 339-341 in the revised manuscript) (clean version).

R2_64: Line 310-311   rephrase

RESPONSE: We have rephrased this line accordingly.

R2_65: Line 311        in contrast? For the growing season ED also decreased

RESPONSE: We have deleted this word in the revised manuscript.

R2_66: Line 319    sediment load (you don't measure the availability)

RESPONSE: We have changed this to "sediment load" in the revised manuscript.

R2_67: Fig 3      what does the box-whiskers represent? Are you sure that the precipitation in May period 2 is correct? It seems unlikely that it will have so little variation as period 1 has the largest variation of all months (and where are the whiskers?).

RESPONSE: We have revised this figure (Please see Figure 3). Still the data suggest that monthly precipitation in May has less variation in Period II. The box-whiskers represent the minimum and the maximum values.

R2_68: Line 324-325          please give a mean annual flow in mm for the different periods

RESPONSE: We have revised this line accordingly. We have added mean annual flow at the beginning of the paragraph "Streamflow in Period I was higher than that of Period II, and the mean annual streamflow was 188 and 146 mm.yr$^{-1}$ for Periods I and II, respectively". (Please see lines 364-365 in the revised manuscript) (clean version)

R2_69: Line 326-327          in fig 4 it looks like both are between 2 and 3 l/s at Q70. Q50 and Q10 also looks wrong. Q10 looks like it is certainly above 10 l/s. Please check.

RESPONSE: Thank you for pointing this out. We have updated this line such as follows "The Q70% low flow of the two periods was 2.7 and 2.4 l.s$^{-1}$, the Q50% median flow was 4.0 and 3.1 l.s$^{-1}$, and the Q10% high flow was 10.7 and 7.5 l.s$^{-1}$, respectively". (Please see lines 367-369) (clean version).

R2_70: Line 327-331      rephrase

RESPONSE: We have rephrase these lines to "The decreased flow regime of Period II, probably partially due to an increased evapotranspiration over the past decades (Duethmann and Blöschl, 2018), indicates that streamflow cannot account for the observed increased sediment load of Period II, otherwise an increased streamflow would be expected in Period II". (Please see lines 370-373) (clean version).

R2_71: Line 324-331          it is surprising that the stream flow is found to decrease as the precipitation is shown to increase. Does the change in evaporation seem credible if evap = precip – runoff, in the two periods? It wouldn't be the first time if either the precipitation timeseries or the Q timeseries or both) were not homogeneous through time and let to bias'es.

RESPONSE: Precipitation in our analysis is not significantly different between Periods I and II. This means, according to Q=P-E, that the decrease in streamflow was partially due to an increase in evapotranspiration, because both arable land and forest land of Period II increased a lot compared to Period I. We also analyzed data on daily temperature changes between 1945 and 2017 and found a considerable increase of 1.5°C for this 72 year period. However, we did not include this into the paper because it is out of the main focus. We intended to point out that the behavior of the streamflow data does not support the hypothesis of an impact of climate change on the change in sediment loads. This is, because for a given environmental setting, usually a correlation between flow rates and sediment concentrations exist. This may also be demonstrated in the HOAL catchment, where annual sediment loads within the

different periods are well correlated with annual flow, which in turn is correlated to annual rainfall.

R2_72: Line 340          grassland decreased to? Are the changes statistically significant?

RESPONSE: Yes, this is correct. Grassland was converted to other land uses (mainly cropland and forest land), and grassland decreased a lot from between 14 to 22 percent of the total area in Period I to between 3 to 4 percent in Period II. We did not test this statistically but could do it if requested by the reviewers.

RESPONSE: To our understanding, scale issues (often addressed by means of connectivity) result from landscape heterogeneity. So, once heterogeneity changes, connectivity would change accordingly. In our case study, both the decrease in crop diversity and the increase in field sizes leads to an increase in landscape connectivity. We therefore think that the loss of heterogeneity (due to different factors such as type of crop, particular erosion risk of crop and increase of field sizes) is the most relevant process to explain the sediment loading change in this catchment.

R2_74: Line 345-346          rephrase, and why. The catchment is very small. Larger fields will in it self lead to less variation, and the variation wont be important in itself (explain if I am wrong), it is the properties of the crops that matters?

RESPONSE:

Generally speaking, larger fields mean less heterogeneity, because field border structures that existed previously will disappear. These structures are very important for connectivity because they tend to slow down surface runoff and retain detached soil particles. However, it is not only the landscape structures that affect sediment movement in a landscape but also the crop types which affects flow velocities and transport capacities of overland flow.

We suggest to rephrase this part into: "This shift to agricultural uniformity is likely to act as a land structure effect. A loss of heterogeneity of crop types increases the probability that different fields are managed with the same crop. Then even smaller fields may behave similar to larger fields in terms of sediment production." (Please see lines 393-395) (clean version)

R2_75: Line 350     what does agricultural industrialization mean in this context

RESPONSE: This refers to the massive application of advanced machinery and fertilization technologies that started right after 1950s. This is a common use term in

studies dealing with changes that took place in agriculture over the last decades. We have added this information to the revised manuscript. (Please see lines 400-401) (clean version).

R2_76: Table 1   please add Parcel (n)

RESPONSE:We have added this to Table 1.

R2_77: Section 3.2        Try to make the results in this section more relevant to the sediment load results. What does the findings mean? What are the physical processes that change as fields becomes larger and how does it affect the sediment load?

RESPONSE: We now explain in more detail how the changes in land use and land structure are relevant for sediment load. Generally speaking, the key process that affected sediment load is the change in "heterogeneity" and the change in "boundary effects". In the revised manuscript, both terms are explicitly mentioned in the text, which, we hope, will be helpful for understanding the analysis. We also refer to R2_74.

R2_78: Line 379-380            as you look at the data on a monthly scale you can't state that there is a faster response to increasing flow.

RESPONSE: According to our analysis, the slope of the SRC (i.e. coefficient $b$) for Period II is greater than for Period I. This means that sediment concentration responded faster to increasing streamflow even when using monthly data.

R2_79: Line 383       delivery instead of availability?

RESPONSE: We prefer "sediment availability" to "delivery". To our understanding, the term "delivery" is more dependent on streamflow flow capacity, whilst "sediment availability" or sediment supply is more associated with the erosion process from hillslope surfaces. Here, we analyzed ED between the two periods, which commonly exerts its influence on the hillslope surface.

R2_80: Line 404        is it log10 or Ln? (statisticians often use log for Ln, which make it difficult for the rest of us..)

RESPONSE: Thank you for the question. Here the log () represents log10. To make the expression clear, we have added an explanation in the caption of Figure 6 of the revised manuscript which now reads as "log (a) in x-axis represents the decadal logarithm. ."

R2_81: Line 422        just land cover, the statistics does not change erosion ðŸ˜Š

RESPONSE: In our analysis, both land use and landscape structure displayed various degrees of change. For example, parcel density of arable land in Period I is 1.7-2.0, it is only 0.3-0.6 in Period II. The percent area of arable land in Period I is 73-82% (varying in different years), while it is almost constant at 81-82% during Period II. Again, parcel density of grassland in Period I is 5.2-7.2, while it is 0.9 in Period II. As for the percent area of grassland, it decreased from 14-22% in Period I to 3-4 % in Period II. All of these statistics are displayed in Table 1, and suggests, that it is reasonable to attribute the shift in sediment load to the change in hydrological connectivity, resulting from both landscape structural changes and land use/cover changes.

R2_82: Fig 7      the text is not very clear in my pdf version.

RESPONSE: We have changed the format of the text in Figure 7.

R2_83: Line 433        at Q70% they are the same, therefore higher not considerably higher at >Q70%

RESPONSE: We agree and we have changed the wording to "Specifically, at a given Q higher than approximately $Q_{70\%}$, sediment concentrations in Period II_G were higher than those in Period I_G" (Please see lines 492-495) (clean version).

R2_84: Line 436        this is a very large increase. Are you sure there are no biases in e.g. the sampling technique? It has been seen that for example ISCO intakes placed at/close to the bottom in small stream yielded much larger values than bottle samples. Have you looked into this?

RESPONSE: We agree that sampling takes place at different positions in a river and that this may be an issue. However, in our case we can assume perfect mixing for the manual sampling as well as the automatic sampling. The sampling site is a very small brook with an average base flow rate of around 2 l/s. Manual sampling as well as automated sampling was taken directly in the mixing zone after the weir, where the water is completely disturbed. Thus, we have no reason to assume that the observed differences are due to a different positioning of the sampling devices.

R2_85: Line 467        the row crops are not erosive they are "prone to erosion" or apply a high degree of erodibility to the soil.

RESPONSE: We have deleted this sentence. .

R2_86: Sec 3.4  I think this need a little more introduction. It is not clear to me how you arrive at the results in fig 8.

RESPONSE: May we refer to Lines 305-333, where the details on how this Figure was developed are described.

R2_87: Line 487      reverse periods

RESPONSE: We have revised this line. We changed the sentence to "We found that sediment load increased substantially from Period I to Period II.

R2_88: Line 489      it needs to be more clear what effects land consolidation has in this particular area, in other corners of the world it will have different effects. In my corner, it probably won't lead to larger fields. Larger fields are only a problem if the terrain is sloping and there are substantial physical barriers restricting flow and erosion/sediment transport between the fields, that will disappear when creating larger fields.

RESPONSE: We agree that the observed effect may be quite different in different environments as well as we do agree with the fact that these issues are a problem of sloping land. However, the references used in our work indicate that this issue is not strictly limited to our study catchment, but some fundamental principles of sediment transport on intensively used agricultural land are revealed. To account for the reviewer's request, we have included the following text in the discussion section of the manuscript "This finding is particularly important in regions, where a strong intensification of agricultural management took place during the last decades. ". (Please see lines 584-586 in the revised manuscript) (clean version).

R2_89: Line 507      Which Qxx% is 15 l/s

RESPONSE: We have added an explanation on this, that is "approximately corresponding to Q13% in Period I and Q4% in Period II, respectively". (Please see line 603 in the revised manuscript) (clean version)

R2_90: Line 526-533      is this a general discussion or related to the results of this paper?

RESPONSE: This is a discussion of the observed effects in our test catchment.

R2_91: Line 549-559      If you want to include these new data it should be included in the results and method chapter. Consider deleting.

RESPONSE: This section is to support to the findings of our measurements. The calculations of the different RUSLE factors (C-Factor and LS-Factor) demonstrate

that the results we measured are also consistent with commonly known erosion principles. We consider the use of the RUSLE model as a very common tool in soil erosion modelling. Because of the widespread use of the RUSLE model, we think that inclusion of a detailed procedure to calculate these factors would not be needed. However, we have added a reference to this part, which describes the methodology that has been used. This now reads as "This finding is supported by estimates of the management factor (C-Factor) and the slope and slope length factor (SL-Factor) of the RUSLE for Period I and Period II (Fiener et al., 2020)". (Please see lines 554-556 in the revised manuscript) (clean version)

Fiener, P., Dostal, T., Krasa, J., Schmaltz, E., Strauss, P., and Wilken, F., 2020: Operational USLE-Based Modelling of Soil Erosion in Czech Republic, Austria, and Bavaria – Differences in Model Adaptation, Parametrization, and Data Availability. Applied sciences, 10, 3647, 1-18. Doi:10.3390/app10103647.

R2_92: Line 566-568          conclude on the findings of the paper. This belongs under discussion.

RESPONSE: We partially agree that this sentence might as well be placed in a different section, but probably better in the summary section. However, such a statement is already part of the summary section (lines 31 ff.). Thus, we suggest to keep it to underline the central message of the manuscript once more.

R2_93: Line 576-577          rephrase

RESPONSE: We have deleted the word "normal". The sentence now reads "Meanwhile, unfavorable land use or land cover change will increase sediment load at most flow conditions, although the relevance of this process would decrease at high or very high flow rates."

---

## Editor Decision (ED1)

**Agricultural intensification vs climate change: What drives long-term changes of sediment load?**

ShengpingWang[1,2,3*], Peter Strauss[2], Carmen Krammer[2], Elmar Schmaltz[2], Borbala Szeles[3,4], Günter Blöschl[3,4]

1. *College of Hydraulic and Hydro-Power Engineering, North China Electric Power University, Beijing, 102206, P.R.China*
2. *Institute for Land and Water Management Research, Federal Agency for Water Management, A-3252 Petzenkirchen, Austria*
3. *Institute of Hydraulic Engineering and Water Resources Management, Vienna University of Technology, Vienna, Austria*
4. *Vienna Doctoral Programme on Water Resource Systems, Vienna University of Technology, Vienna, Austria*

**\*Corresponding author**: Shengping Wang

Email: wangshp418@126.com;Shengping_Wang@ncepu.edu.cn

**Abstract:** Climate change and agricultural intensification are expected to increase soil erosion and sediment production from arable land in many regions. However, so far, most studies were based on short-term monitoring and/or modeling, making it difficult to assess their reliability in terms of estimating long-term changes. We present the results from a unique data set consisting of measurements of sediment loads from a 60ha catchment (the Hydrological Open Air Laboratory, HOAL, in Petzenkirchen, Austria) over a time period spanning 72 years. Specifically, we compare Period I (1946-1954) and Period II (2002-2017) by fitting sediment rating curves for the growth and dormant seasons for each of the periods. The results suggest a significant increase in sediment loads from Period I to Period II with an average of

删除[胡杨 [2]]: commonly took land use/cover change and landscape structure change as a whole when perofrming such attribution analysis, and most have been

删除[胡杨 [2]]: in

删除[胡杨 [2]]: window

删除[胡杨]: yield

$11.6\pm10.8$ ton·yr⁻¹ to $63.6\pm84.0$ ton·yr⁻¹. The sediment flux changed mainly due to a shift of the sediment rating curves (SRC), given that the mean daily discharge significantly decreased from $5.0\pm14.5$ l·s⁻¹ for Period I to $3.8\pm6.6$ l·s⁻¹ for Period II.

30 The slopes of the SRC for the growing season and the dormant season of Period I were 0.3 and 0.8, respectively, whilst they were 1.6 and 1.7 for Period II, respectively. Climate change, considered in terms of rainfall erosivity, was not responsible for this shift, given that erosivity decreased by 30.4% from the dormant season of Period I to that of Period II, and no significant difference was found between the growing

35 seasons of Periods I and II. However, the change in sediment flux can be explained by the changes in crop type and parcel structure. At low and median streamflow conditions, land consolidation in Period II (*i.e.* the parcel effect) had no apparent influence on sediment production. Whilst with increasing stream flow, parcel structure became more important in controlling sediment yield, as a result of an enhanced

40 sediment connectivity in the landscape, leading to a dominant role at high flow conditions. The increase in cropland between Periods I and II at the expense of grassland led to an increase in sediment flux, although it's relevance was surpassed by the effect of parcel structure change at high flow condition. We conclude that both land cover change and land consolidation should be accounted for when assessing

45 sediment flux changes. Especially during high flow conditions, parcel structure change substantially alters sediment fluxes, which is most relevant for long-term sediment loads and land degradation. Increased attention to improving parcel structure is therefore needed in climate adaptation and agricultural catchment management.

删除[胡杨]: annual

删除[胡杨 [2]]: streamflow

删除[胡杨]: varied

设置格式[胡杨 [2]]: 字体: 非倾斜

设置格式[胡杨 [2]]: 字体: 非倾斜

删除[胡杨]: little between the periods (5.6 *ls⁻¹* and 7.6 *ls* …

删除[胡杨]: the log regression lines of

删除[胡杨]: 16.72

删除[胡杨]: 4.9

删除[胡杨]: 5.38

删除[胡杨]: 17

删除[胡杨]: changes

删除[胡杨]: During

删除[胡杨]: (*i.e.* $Q < Q_{20\%}$)

删除[胡杨]: did not exert

删除[胡杨]: an

删除[胡杨]: ($Q > Q_{20\%}$)

删除[胡杨]: played an

删除[胡杨]: increasingly role

删除[胡杨]: contribution

删除[胡杨]: and

删除[胡杨]: due to enhanced sediment connectivity in the …

删除[胡杨]: extremely

删除[胡杨 [2]]: (*i.e.* $Q > Q_{2\%}$)

删除[胡杨]: in

删除[胡杨]: had an unfavourable effect on

删除[胡杨]: , independent of streamflow

删除[STRAUSS Peter]: change's

删除[胡杨 [2]]: effect a

删除[胡杨]: , with declining relevance as flow increased

删除[胡杨 [2]]: simultaneously

删除[胡杨]: extremely

删除[胡杨]: land consolidation

[revised manuscript text omitted]

设置格式[STRAUSS Peter]: 字体: 五号, 无下划线, 字体颜色: 自动设置, 德语(奥地利)

删除[胡杨]: 2021.

删除[胡杨]: .

删除[胡杨]: .

设置格式[胡杨]: 字体: 五号

---

## Author Response (AR2)

* * *
*The following documents the comments from the editors/reviewers and the response made by the authors to the comments one by one. The comments from the editors/reviewers are displayed in black, while the response made by the authors are shown in blue.*
* * *
Dear Dr. Wang,

C#1:Thank you for the submission of the revised version of your manuscript.

You replied to most comments by the reviewers in a comprehensive manner and addressed the criticized aspects and implemented suggested improvements.

When reading the revised version, I still noticed some linguistic issues such as a number of very long sentences that are better split in two or formulations that may be misleading. I annotated your track-changes documents with a number of such observations for suggesting improvements.

Response: Thank you for the suggestions/comments to our manuscript. We really appreciate the time and energy the editors have invested in our manuscript. We have revised our manuscript by taking into account the suggestions/comments raised by the editors/reviewers. We hope that the changes made improved our paper and the quality of our manuscript meets the requirement of the journal.

C#2 There is one remaining issue related to the results and their interpretation. This is the question of how the different sampling regimes during the two periods might have affected your findings. Despite the fact that you discuss that aspect and provide qualitative arguments why you expect the methodological effect to be small, I am not fully satisfied and think you have not yet fully explored the possibilities to get some detailed insights. Given the fact that you have a more advanced sampling regime for the second period, your SRC according to Eq. [3] should yield a fairly accurate representation of Period II. This SRC allows to calculate a theoretical, predicted sediment concentration time series for Period II (based on the continuous discharge data). Knowing the sampling scheme of Period I, you can resample in a Monte Carlo fashion this theoretical concentration time series and derive the respective SRC.

Of course, this is not a full substitute for measured data during Period I at higher frequencies, but such a resampling allows for a consistent assessment evaluation of what could have been observed during Period II. Such a quantitative estimate of the effect of the sampling schemes is important (as highlighted by Rev. 1) especially given the situation that your loads are dominated by a few events (8 events in Period II, 25 in Period I). With grab sampling these events will be often only poorly covered and therefore it is essential to know to the best possible degree whether the sampling has an impact on the resulting SRCs.

Response: We agree with this comment. We have changed the methodology to derive

sediment loads. We are now using a combination of SRC together with the continuous flow data that are available for both periods. This has the advantage of being more stable with respect to different sampling scheme frequencies (Thomas, 1988; Syvitski et al., 2000; Desilets et al., 2007; Sheridan et al., 2011). Thus, we have revised the respective section (2.3.2 Sediment regime analysis) in the manuscript. We became aware that the information on availability of high resolution flow data for both periods may have not been indicated sufficiently. We have therefore slightly modified section '2.2 Data availability' to make this clear. As a result, the calculated sediment loads for Period I changed from around 11 t yr$^{-1}$ to around 6 t yr$^{-1}$.

For Period II we followed the suggestion of the reviewer and carried out a resampling. Considering that Period II has a finer-resolution dataset, we resampled the dataset of Period II (random selection of the same number of samples as available in Period I, which was then repeated 10 times) and compared the results between the original data set (number of samples n= 5175) and the resampled data set (number of samples n= 935 ) of Period II.

We fitted a sediment rating curve for each resampled data set and then estimated sediment load for Period II by combining the resampled SRC with the continuously available flow data. We found that the mean annual sediment load of Period II, derived from the resampling with adapted numbers of sampling points ($62.4 \pm 10.2$ ton.yr$^{-1}$) was not significantly different from our original calculations ($60.0 \pm 140.0$ ton.yr$^{-1}$). Although this is a rough analysis based on only 10 resamplings, the results suggest that an altered sampling frequency has no significant influence on sediment load in our case study.

We added this information to section '4.2 Potential interference of different sampling methods'.

C#3 Please have a look at my comments in the annotated manuscript and incorporate the re-sampling results to back up (or modify) your findings and their interpretation.

Response: Thanks for all of the valuable comments and suggestions, which, we believe, are greatly beneficial to improving our manuscript. We have revised the long sentences annotated in the manuscript. We also followed the other suggestions and made responses to the comments one by one as follows.

C#4    Line 37 "theparcel"?
Response: we have changed to "the parcel".

C#5    Line 65, "Zhang et al. (2021) quantitatively evaluated..." So climate change reduced erosion risk? What kind of land use change had happened? Without such information, the following numbers are not informative.
Response: Thank you for the question. We have revised the expression as "Zhang et al. (2021) quantitatively evaluated the contributions of the decrease in annual rainfall erosivity, the decrease in arable land and bare land, and the construction of silt trap

dams to the reduction of sediment load of a typical Loess watershed". (Please see line 67 in the revised manuscipt)

C#6 Line 70, "management practices had a greater impact on erosion than climate change" Again, the information is too limited.
Response: We have changed it as "They found that the conservational management practices had a greater impact on reducing soil erosion rates than forecasted effects of climate change (i.e. the decrease in rainfall amounts in erosion sensitive months)". (Please see line 73 in the revised manuscript)

C#7 Line 76-83. Long sentence.
Response: Thanks you for pointing this out. We have rephrased these lines. (Please see line 80-85 in the revised manuscript)

C#8 Line 101-104: Could also be split into two sentences.
Response: Thank you for pointing this out. We have split these sentences into two.(Please see line 106-111 in the revised manuscript)

C#9 Line 111, State already here that these sediment measurements were carried out only during specific periods. As written, the text gives the impression that there is a continuous observational data set (triggering comments such as by rev. 2).
Response: we have revised it as "have been monitored in the HOAL catchment from 1945 to 1954 and from 2000 to now" (Please see line 119 in the revised manuscript). We have also added the word 'periodically' into the section 'Abstract' to make this clear. Additionally we have added some sentences into the section '2.2 Data availability' as follows:

"A data set of discharge and sediment concentration was available for the period 1945-1954. After that, measurements were stopped and started again in 1990. Therefore, data records for the period 1946-1954 (Period I) and 2002-2017 (Period II) were used for this analysis."

C#10 Line 115,You use the term land use with different meanings in the same sentence. Define clearly what the term means and use it in a consistent manner throughout the manuscript.
Response: We are grateful for this comment. We went thoroughly through the manuscript and replaced as much of the 'land use' 'land cover' wording with the term LUCC. We recognized that we did not define the terminology properly so far. Therefore, we added sentences to define LUCC to the section 'Introduction' as follows:
'In our analysis, we evaluate the relative roles of climate change, LUCC and the change of land structure on sediment production. We define LUCC as a change in either type of land use (i.e. arable land, grassland, forest) or type of land cover

(agricultural management, mainly by crops with different risk of soil erosion).'(Please see line 107-109 in the revised manuscript)

C#11  Line 136: insert periodically
Response: Thank you for pointing this out. We have inserted "periodically".

C#12 Line 141-142:This statement should go into the discussion, not the method section. It is irrelevant for understanding what you did.
Response: we have removed the whole sentence, i.e. "To our knowledge, for the time period 1945-1954, almost no sediment concentration data 140 are available in Austria, we therefore think that this database from the HOAL is extremely valuable and relevant for climate impact analysis".

C#13 Line 226-230, Isn't it possible to resample Period II with the sampling scheme of Period I? This would provide additional information how robust the comparison actually is.
Response: We are grateful for this suggestion. We have resampled the dataset of Period II to account for the different amount of samples in Period I, and carried out an analysis to test whether the influence of sampling scheme is relevant when assessing sediment regime in our case study. For detailed explanation we refer to C#2.

C#14 Line 240, What were the covariates you considered?
Response: We carried out this analysis by considering the log-transformed discharge (log_Q) as independent variable, the different periods as the fixed factor (Period), and the log-transformed sediment concentration (log_C) as the dependent variable. This analysis was implemented in SPSS (General linear model---Univariate---model). By examining the interaction effect between the independent variable and the fixed factor, we tested if the slopes of the regression lines of log_C against log_Q were significantly different between the periods. In the revised manuscript, we revised the text as "by ANCOVA analysis with the log-transformed discharge as independent variable".

C#15 Line 360, Can you provide numbers?
Response: We have added the amount of snowfall in the revised manuscript (about 10%).

C#16 Line 403, Parcel density can be skipped: this is redundant information.
Response: We agree that it is a redundant information. However, we think that, it might be easier for the readers to directly know this number without a necessity to calculate it. So, we suggest to keep the text as before.

C#17 Line 468, What is shifted sediment regime? Not clear.
Response: Thanks for the question. According to Asselman (2000), the plot of slope

against intercepts (i.e. a/b) is representative of a sediment regime of a river section or a catchment. The values of the regression coefficients of the sediment rating curves, i.e. *a* and *b*, depend on the erosion severity, the availability of sediment in a certain area, the power of the river to erode and transport the available material, and on the extent to which new sediment sources become available during weather conditions that cause high discharge. High values of the coefficient 'a' occur in areas characterized by intensively weathered materials, which can easily be eroded and transported. The *b* coefficient represents the erosive power of the river (Peters-Kummerly, 1973; Morgan, 1995). Therefore, the shift of sediment regimes means an alteration of either soil erodibility and/or erosive power of the river. The explanation for this shift is indicated in the text directly below figure 6. In the revised manuscript, we indicated this with the addition '(see text below)'.

"According to Asselman (2000), a shift of sediment regime means an alteration of either soil erodibility and/or erosive power of the river. In Figure 6, we found that the regression lines of Periods I to II are different."(Please see line 480 in the revised manuscript).

C#18 Line 475, Effect if sampling method?
Response: We have carried out a resampling analysis, according to the suggestion of the editors/reviewers, to test whether different sampling schemes had effects on sediment load estimation. We found that sediment loads estimation were not different between the resampled data set and the original sampling scheme. For details please see C#2. Therefore, according to the physical meaning of the plot of *a/b,* we mainly attributed the shift of the line to the alteration of sediment transport regime, instead of the interference of the sampling schemes.

C#19 Line 573-574, Not convincing: different sampling strategies on non-linear will cause different average concentrations.
Response: We agree that this statement only provides a qualitative argument which is indeed not convincing. To answer this comment, in the revised manuscript we deleted this statement.

---

## Author Response (AR3)

**Response to the editors**

Dear editors, we highly appreciate that our manuscript now is ready for acceptance and was suggested as a highlight paper by the editors. We are very grateful for the time and the energy you have spent to our manuscript. We also would like to thank the reviewers for their suggestions which – we believe – improved the manuscript to a large extent.

Base on your last comments we have discussed the issue of authorship. We decided to follow your suggestion for change of senior authorship. We also rearranged the appearing position of different authors and added more specific information to make the different contributions clearer. There is one more thing regarding authorship: We have now included Zhiqiang Zhang as co-author, which we had – due to editorial error forgotten to include before. This may seem to be an unfortunate twist, however we can assure you that Zhiqiang Zhang contributed substantially to the manuscript. He reviewed the first draft of the manuscript and contributed to the revision by reviewing the reply to the comments and editing the text (see text on personal contributions). We hope this new arrangement is accepted by HESS.

Best regards,
Shengping Wang